# Efficient Conditionally Invariant Representation Learning

**Roman Pogodin**[*]
Gatsby Unit, UCL
rmn.pogodin@gmail.com

**Namrata Deka**[*]
UBC
dnamrata@cs.ubc.ca

**Yazhe Li**[*]
DeepMind & Gatsby Unit, UCL
yazhe@google.com

**Danica J. Sutherland**
UBC & Amii
dsuth@cs.ubc.ca

**Victor Veitch**
UChicago & Google Brain
victorveitch@google.com

**Arthur Gretton**
Gatsby Unit, UCL
arthur.gretton@gmail.com

## Abstract

We introduce the Conditional Independence Regression CovariancE (CIRCE), a measure of conditional independence for multivariate continuous-valued variables. CIRCE applies as a regularizer in settings where we wish to learn neural features $\varphi(X)$ of data $X$ to estimate a target $Y$, while being conditionally independent of a distractor $Z$ given $Y$. Both $Z$ and $Y$ are assumed to be continuous-valued but relatively low dimensional, whereas $X$ and its features may be complex and high dimensional. Relevant settings include domain-invariant learning, fairness, and causal learning. The procedure requires just a single ridge regression from $Y$ to kernelized features of $Z$, which can be done in advance. It is then only necessary to enforce independence of $\varphi(X)$ from residuals of this regression, which is possible with attractive estimation properties and consistency guarantees. By contrast, earlier measures of conditional feature dependence require multiple regressions for each step of feature learning, resulting in more severe bias and variance, and greater computational cost. When sufficiently rich features are used, we establish that CIRCE is zero if and only if $\varphi(X) \perp\!\!\!\perp Z \mid Y$. In experiments, we show superior performance to previous methods on challenging benchmarks, including learning conditionally invariant image features.

## 1 Introduction

We consider a learning setting where we have labels $Y$ that we would like to predict from features $X$, and we additionally observe some metadata $Z$ that we would like our prediction to be 'invariant' to. In particular, our aim is to learn a representation function $\varphi$ for the features such that $\varphi(X) \perp\!\!\!\perp Z \mid Y$. There are at least three motivating settings where this task arises.

1. Fairness. In this context, $Z$ is some protected attribute (e.g., race or sex) and the condition $\varphi(X) \perp\!\!\!\perp Z \mid Y$ is the equalized odds condition (Mehrabi et al., 2021).

2. Domain invariant learning. In this case, $Z$ is a label for the environment in which the data was collected (e.g., if we collect data from multiple hospitals, $Z_i$ labels the hospital that the $i$th datapoint is from). The condition $\varphi(X) \perp\!\!\!\perp Z \mid Y$ is sometimes used as a target for invariant learning (e.g., Long et al., 2018; Tachet des Combes et al., 2020; Goel et al., 2021; Jiang & Veitch, 2022). Wang & Veitch (2022) argue that this condition is well-motivated in cases where $Y$ causes $X$.

3. Causal representation learning. Neural networks may learn undesirable "shortcuts" for their tasks – e.g., classifying images based on the texture of the background. To mitigate this issue, various schemes have been proposed to force the network to use causally relevant factors in its decision (e.g., Veitch et al., 2021; Makar et al., 2022; Puli et al., 2022). The structural causal assumptions used in such approaches imply conditional independence relationships between the features we would like the network to use, and observed metadata

---

[*]Equal contribution. [†]Code for image data experiments is available at github.com/namratadeka/circe

that we may wish to be invariant to. These approaches then try to learn causally structured representations by enforcing this conditional independence in a learned representation.

In this paper, we will be largely agnostic to the motivating application, instead concerning ourselves with how to learn a representation $\varphi$ that satisfies the target condition. Our interest is in the (common) case where $X$ is some high-dimensional structured data – e.g., text, images, or video – and we would like to model the relationship between $X$ and (the relatively low-dimensional) $Y, Z$ using a neural network representation $\varphi(X)$. There are a number of existing techniques for learning conditionally invariant representations using neural networks (e.g., in all the motivating applications mentioned above). Usually, however, they rely on the labels $Y$ being categorical with a small number of categories. We develop a method for conditionally invariant representation learning that is effective even when the labels $Y$ and attributes $Z$ are continuous or moderately high-dimensional.

To understand the challenge, it is helpful to contrast with the task of learning a representation $\varphi$ satisfying the marginal independence $\varphi(X) \perp\!\!\!\perp Z$. To accomplish this, we might define a neural network to predict $Y$ in the usual manner, interpret the penultimate layer as the representation $\varphi$, and then add a regularization term that penalizes some measure of dependence between $\varphi(X)$ and $Z$. As $\varphi$ changes at each step, we'd typically compute an estimate based on the samples in each mini-batch (e.g., Beutel et al., 2019; Veitch et al., 2021). The challenge for extending this procedure to conditional invariance is simply that it's considerably harder to measure. More precisely, as conditioning on $Y$ "splits" the available data,[1] we require large samples to assess conditional independence. When regularizing neural network training, however, we only have the samples available in each mini-batch: often not enough for a reliable estimate.

The main contribution of this paper is a technique that reduces the problem of learning a conditionally independent representation to the problem of learning a marginally independent representation, following a characterization of conditional independence due to Daudin (1980). We first construct a particular statistic $\zeta(Y, Z)$ such that enforcing the marginal independence $\varphi(X) \perp\!\!\!\perp \zeta(Y, Z)$ is (approximately) equivalent to enforcing $\varphi(X) \perp\!\!\!\perp Z \mid Y$. The construction is straightforward: given a fixed feature map $\psi(Y, Z)$ on $\mathcal{Y} \times \mathcal{Z}$ (which may be a kernel or random Fourier feature map), we define $\zeta(Y, Z)$ as the conditionally centered features, $\zeta(Y, Z) = \psi(Y, Z) - \mathbb{E}[\psi(Y, Z) \mid Y]$. We obtain a measure of conditional independence, the *Conditional Independence Regression CovariancE* (CIRCE), as the Hilbert-Schmidt Norm of the kernel covariance between $\varphi(X)$ and $\zeta(Y, Z)$. A key point is that the conditional feature mean $\mathbb{E}[\psi(Y, Z) \mid Y]$ can be estimated offline, in advance of any neural network training, using standard methods (Song et al., 2009; Grunewalder et al., 2012; Park & Muandet, 2020; Li et al., 2022). This makes CIRCE a suitable regularizer for any setting where the conditional independence relation $\varphi(X) \perp\!\!\!\perp Z \mid Y$ should be enforced when learning $\varphi(X)$. In particular, the learned relationship between $Z$ and $Y$ doesn't depend on the mini-batch size, sidestepping the tension between small mini-batches and the need for large samples to estimate conditional dependence. Moreover, when sufficiently expressive features (those corresponding to a *characteristic kernel*) are employed, then CIRCE is zero if and only if $\varphi(X) \perp\!\!\!\perp Z \mid Y$: this result may be of broader interest, for instance in causal structure learning Zhang et al. (2011) and hypothesis testing Fukumizu et al. (2008); Shah & Peters (2020); Huang et al. (2022).

Our paper proceeds as follows: in Section 2, we introduce the relevant characterization of conditional independence from (Daudin, 1980), followed by our CIRCE criterion – we establish that CIRCE is indeed a measure of conditional independence, and provide a consistent empirical estimate with finite sample guarantees. Next, in Section 3, we review alternative measures of conditional dependence. Finally, in Section 4, we demonstrate CIRCE in two practical settings: a series of counterfactual invariance benchmarks due to Quinzan et al. (2022), and image data extraction tasks on which a "cheat" variable is observed during training.

## 2 EFFICIENT CONDITIONAL INDEPENDENCE REGULARIZER

We begin by providing a general-purpose characterization of conditional independence. We then introduce CIRCE, a conditional independence criterion based on this characterization, which is zero if and only if conditional independence holds (under certain required conditions). We provide a finite sample estimate with convergence guarantees, and strategies for efficient estimation from data.

---

[1]If $Y$ is categorical, naively we would measure a marginal independence for each level of $Y$.

## 2.1 Conditional independence

We begin with a natural definition of conditional independence for real random variables:

**Definition 2.1** (Daudin, 1980). $X$ and $Z$ are $Y$-conditionally independent, $X \perp\!\!\!\perp Z \mid Y$, if for all test functions $g \in L^2_{XY}$ and $h \in L^2_{ZY}$, i.e. for all square-integrable functions of $(X, Y)$ and $(Z, Y)$ respectively, we have almost surely in $Y$ that

$$\mathbb{E}_{XZ}\left[g(X,Y)\,h(Z,Y)\,|\,Y\right] = \mathbb{E}_X\left[g(X,Y)\,|\,Y\right]\mathbb{E}_Z\left[h(Z,Y)\,|\,Y\right]. \tag{1}$$

The following classic result provides an equivalent formulation:

**Proposition 2.2** (Daudin, 1980). *$X$ and $Z$ are $Y$-conditionally independent if and only if it holds for all test functions $g \in E_1 = \left\{g \in L^2_{XY} \mid \mathbb{E}_X\left[g(X,Y)\,|\,Y\right] = 0\right\}$ and $h \in E_2 = \left\{h \in L^2_{ZY} \mid \mathbb{E}_Z\left[h(Z,Y)\,|\,Y\right] = 0\right\}$ that*

$$\mathbb{E}[g(X,Y)\,h(Z,Y)] = 0. \tag{2}$$

Daudin (1980) notes that this condition can be further simplified (see Corollary A.3 for a proof):

**Proposition 2.3** (Equation 3.9 of Daudin 1980). *$X$ and $Z$ are $Y$-conditionally independent if and only if it holds for all $g \in L^2_X$ and $h \in E_2 = \left\{h \in L^2_{ZY} \mid \mathbb{E}_Z\left[h(Z,Y)\,|\,Y\right] = 0\right\}$ that*

$$\mathbb{E}[g(X)\,h(Z,Y)] = 0. \tag{3}$$

An equivalent way of writing this last condition (see Lemma B.1 for a formal proof) is:

$$\text{for all } g \in L^2_X \text{ and } h \in L^2_{ZY}, \quad \mathbb{E}\left[g(X)\left(h(Z,Y) - \mathbb{E}_{Z'}\left[h(Z',Y)\,|\,Y\right]\right)\right] = 0. \tag{4}$$

The reduction to $g$ not depending on $Y$ is crucial for our method: when we are learning the representation $\varphi(X)$, then evaluating the conditional expectations $\mathbb{E}_X\left[g(\varphi(X),Y)\,|\,Y\right]$ from Proposition 2.2 on every minibatch in gradient descent requires impractically many samples, but $\mathbb{E}_Z\left[h(Z,Y)\,|\,Y\right]$ does not depend on $X$ and so can be pre-computed before training the network.

## 2.2 Conditional Independence Regression CovariancE (CIRCE)

The characterization (4) of conditional independence is still impractical, as it requires checking all pairs of square-integrable functions $g$ and $h$. We will now transform this condition into an easy-to-estimate measure that characterizes conditional independence, using kernel methods.

A *kernel* $k(x, x')$ is a symmetric positive-definite function $k : \mathcal{X} \times \mathcal{X} \to \mathbb{R}$. A kernel can be represented as an inner product $k(x, x') = \langle \phi(x), \phi(x') \rangle_{\mathcal{H}}$ for a *feature vector* $\phi(x) \in \mathcal{H}$, where $\mathcal{H}$ is a reproducing kernel Hilbert space (RKHS). These are spaces $\mathcal{H}$ of functions $f : \mathcal{X} \to \mathbb{R}$, with the key *reproducing property* $\langle \phi(x), f \rangle_{\mathcal{H}} = f(x)$ for any $f \in \mathcal{H}$. For $M$ points we denote $K_X$. a row vector of $\phi(x_i)$, such that $K_{Xx}$ is an $M \times 1$ matrix with $k(x_i, x)$ entries and $K_{XX}$ is an $M \times M$ matrix with $k(x_i, x_j)$ entries. For two separable Hilbert spaces $\mathcal{G}, \mathcal{F}$, a Hilbert-Schmidt operator $A : \mathcal{G} \to \mathcal{F}$ is a linear operator with a finite Hilbert-Schmidt norm

$$\|A\|^2_{\mathrm{HS}(\mathcal{G},\mathcal{F})} = \sum\nolimits_{j \in J} \|Ag_j\|^2_{\mathcal{F}}, \tag{5}$$

where $\{g_j\}_{j \in J}$ is an orthonormal basis of $\mathcal{G}$ (for finite-dimensional Euclidean spaces, obtained from a linear kernel, $A$ is just a matrix and $\|A\|_{\mathrm{HS}}$ its Frobenius norm). The Hilbert space $\mathrm{HS}(\mathcal{G}, \mathcal{F})$ includes in particular the rank-one operators $\psi \otimes \phi$ for $\psi \in \mathcal{F}, \phi \in \mathcal{G}$, representing outer products,

$$[\psi \otimes \phi]g = \psi \langle \phi, g \rangle_{\mathcal{G}}, \qquad \langle A, \psi \otimes \phi \rangle_{\mathrm{HS}(\mathcal{G},\mathcal{F})} = \langle \psi, A\phi \rangle_{\mathcal{F}}. \tag{6}$$

See Gretton (2022, Lecture 5) for further details.

We next introduce a kernelized operator which (for RKHS functions $g$ and $h$) reproduces the condition in (4), which we call the Conditional Independence Regression CovariancE (CIRCE).

**Definition 2.4** (CIRCE operator). Let $\mathcal{G}$ be an RKHS with feature map $\phi : \mathcal{X} \to \mathcal{G}$, and $\mathcal{F}$ an RKHS with feature map $\psi : (\mathcal{Z} \times \mathcal{Y}) \to \mathcal{F}$, with both kernels bounded: $\sup_x \|\phi(x)\| < \infty$, $\sup_{z,y} \|\psi(z,y)\| < \infty$. Let $X, Y$, and $Z$ be random variables taking values in $\mathcal{X}, \mathcal{Y}$, and $\mathcal{Z}$ respectively. The *CIRCE operator* is

$$C^c_{XZ|Y} = \mathbb{E}\left[\phi(X) \otimes \left(\psi(Z,Y) - \mathbb{E}_{Z'}\left[\psi(Z',Y)\,|\,Y\right]\right)\right] \in \mathrm{HS}(\mathcal{G},\mathcal{F}). \tag{7}$$

For any two functions $g \in \mathcal{G}$ and $h \in \mathcal{F}$, Definition 2.4 gives rise to the same expression as in (4),

$$\left\langle C^c_{XZ|Y}, g \otimes h \right\rangle_{\text{HS}} = \mathbb{E}\left[g(X)\left(h(Z, Y) - \mathbb{E}_{Z'}\left[h(Z', Y) \,|\, Y\right]\right)\right]. \tag{8}$$

The assumption that the kernels are bounded in Definition 2.4 guarantees Bochner integrability (Steinwart & Christmann, 2008, Def. A.5.20), which allows us to exchange expectations with inner products as above: the argument is identical to that of Gretton (2022, Lecture 5) for the case of the unconditional feature covariance. For unbounded kernels, Bochner integrability can still hold under appropriate conditions on the distributions over which we take expectations, e.g. a linear kernel works if the mean exists, and energy distance kernels may have well-defined feature (conditional) covariances when relevant moments exist (Sejdinovic et al., 2013).

Our goal now is to define a kernel statistic which is zero iff the CIRCE operator $C^c_{XZ|Y}$ is zero. One option would be to seek the functions, subject to a bound such as $\|g\|_{\mathcal{G}} \leq 1$ and $\|f\|_{\mathcal{F}} \leq 1$, that maximize (8); this would correspond to computing the largest singular value of $C^c_{XZ|Y}$. For unconditional covariances, the equivalent statistic corresponds to the Constrained Covariance, whose computation requires solving an eigenvalue problem (e.g. Gretton et al., 2005a, Lemma 3). We instead follow the same procedure as for unconditional kernel dependence measures, and replace the spectral norm with the Hilbert-Schmidt norm (Gretton et al., 2005b): both are zero when $C^c_{XZ|Y}$ is zero, but as we will see in Section 2.3 below, the Hilbert-Schmidt norm has a simple closed-form empirical expression, requiring no optimization.

Next, we show that for rich enough RKHSes $\mathcal{G}, \mathcal{F}$ (including, for instance, those with a Gaussian kernel), the Hilbert-Schmidt norm of $C^c_{XZ|Y}$ characterizes conditional independence.

**Theorem 2.5.** *For $\mathcal{G}$ and $\mathcal{F}$ with $L^2$-universal kernels (see, e.g., Sriperumbudur et al., 2011),*

$$\|C^c_{XZ|Y}\|_{\text{HS}} = 0 \quad \text{if and only if} \quad X \perp\!\!\!\perp Z \mid Y. \tag{9}$$

The "if" direction is immediate from the definition of $C^c_{XZ|Y}$. The "only if" direction uses the fact that the RKHS is dense in $L^2$, and therefore if (8) is zero for all RKHS elements, it must be zero for all $L^2$ functions. See Appendix B for the proof. Therefore, minimizing an empirical estimate of $\|C^c_{XZ|Y}\|_{\text{HS}}$ will approximately enforce the conditional independence we need.

**Definition 2.6.** For convenience, we define $\text{CIRCE}(X, Z, Y) = \|C^c_{XZ|Y}\|^2_{\text{HS}}$.

In the next two sections, we construct a differentiable estimator of this quantity from samples.

### 2.3 EMPIRICAL CIRCE ESTIMATE AND ITS USE AS A CONDITIONAL INDEPENDENCE REGULARIZER

To estimate CIRCE, we first need to estimate the conditional expectation $\mu_{ZY|Y}(y) = \mathbb{E}_Z\left[\psi(Z, y) \,|\, Y = y\right]$. We define[2] $\psi(Z, Y) = \psi(Z) \otimes \psi(Y)$, which for radial basis kernels (e.g. Gaussian, Laplace) is $L_2$-universal for $(Z, Y)$.[3] Therefore, $\mu_{ZY|Y}(y) = \mathbb{E}_Z\left[\psi(Z) \,|\, Y = y\right] \otimes \psi(y) = \mu_{Z|Y}(y) \otimes \psi(y)$. The CIRCE operator can be written as

$$C^c_{XZ|Y} = \mathbb{E}\left[\phi(X) \otimes \psi(Y) \otimes \left(\psi(Z) - \mu_{Z|Y}(Y)\right)\right] \tag{10}$$

We need two datasets to compute the estimator: a holdout set of size $M$ used to estimate conditional expectations, and the main set of size $B$ (e.g., a mini-batch). The holdout dataset is used to estimate conditional expectation $\mu_{ZY|Y}$ with kernel ridge regression. This requires choosing the ridge parameter $\lambda$ and the kernel parameters for $Y$. We obtain both of these using leave-one-out cross-validation; we derive a closed form expression for the error by generalizing the result of Bachmann et al. (2022) to RKHS-valued "labels" for regression (see Theorem C.1).

---

[2] We abuse notation in using $\psi$ to denote feature maps of $(Y, Z)$, $Y$, and $Z$; in other words, we use the argument of the feature map to specify the feature space, to simplify notation.

[3] Fukumizu et al. (2008, Section 2.2) show this kernel is *characteristic*, and Sriperumbudur et al. (2011, Figure 1 (3)) that being characteristic implies $L_2$ universality in this case.

The following theorem defines an empirical estimator of the Hilbert-Schmidt norm of the empirical CIRCE operator, and establishes the consistency of this statistic as the number of training samples $B$, $M$ increases. The proof and a formal description of the conditions may be found in Appendix C.2

**Theorem 2.7.** *The following estimator of CIRCE for $B$ points and $M$ holdout points (for the conditional expectation):*

$$\widehat{CIRCE} = \frac{1}{B(B-1)} \mathrm{Tr} \left( K_{XX} \left( K_{YY} \odot \hat{K}_{ZZ}^c \right) \right) . \tag{11}$$

*converges as $O_p(1/\sqrt{B} + 1/M^{(\beta-1)/(2(\beta+p))})$, when the regression in Equation (30) is well-specified. $K_{XX}$ and $K_{YY}$ are kernel matrices of $X$ and $Y$; elements of $K_{ZZ}^c$ are defined as $K_{zz'}^c = \langle \psi(z) - \mu_{Z|Y}(y), \psi(z') - \mu_{Z|Y}(y') \rangle$; $\beta \in (1, 2]$ characterizes how well-specified the solution is and $p \in (0, 1]$ describes the eigenvalue decay rate of the covariance operator over $Y$.*

The notation $O_p(A)$ roughly states that with any constant probability, the estimator is $O(A)$.

**Remark.** For the smoothly well-specified case we have $\beta = 2$, and for a Gaussian kernel $p$ is arbitrarily close to zero, giving a rate $O_p(1/\sqrt{B} + 1/M^{1/4})$. The $1/M^{1/4}$ rate comes from conditional expectation estimation, where it is minimax-optimal for the well-specified case (Li et al., 2022). Using kernels whose eigenvalues decay slower than the Gaussian's would slow the convergence rate (see Li et al., 2022, Theorem 2).

The algorithm is summarized in Algorithm 2. We can further improve the computational complexity for large training sets with random Fourier features (Rahimi & Recht, 2007); see Appendix D.

---

**Algorithm 1** Estimation of CIRCE

Holdout data $\{(z_i, y_i)\}_{i=1}^M$, mini-batch $\{(x_i, z_i, y_i)\}_{i=1}^B$
**Holdout data**
Leave-one-out (Theorem C.1) for $\lambda$ (ridge parameter) and $\sigma_y$ (parameters of $Y$ kernel):
$\lambda, \sigma_y = \arg\min \sum_{i=1}^M \frac{\left\| \psi(z_i) - K_{y_i Y}(K_{YY} + \lambda I)^{-1} K_{Z \cdot} \right\|_{\mathcal{H}_z}^2}{\left( 1 - \left( K_{YY}(K_{YY} + \lambda I)^{-1} \right)_{ii} \right)^2}$
$W_1 = (K_{YY} + \lambda I)^{-1}$, $W_2 = W_1 K_{ZZ} W_1$
**Mini-batch**
Compute kernel matrices $K_{xx}, K_{yy}, K_{yY}, K_{yZ}$ ($x, y, z$: mini-batch, $Y, Z$: holdout)
$\hat{K}^c = K_{yy} \odot \left( K_{zz} - K_{yY} W_1 K_{Zz} - (K_{yY} W_1 K_{Zz})^\top + K_{yY} W_2 K_{Yy} \right)$
$\mathrm{CIRCE} = \frac{1}{B(B-1)} \mathrm{Tr} \left( K_{xx} \hat{K}^c \right)$

---

We can use of our empirical CIRCE as a regularizer for conditionally independent regularization learning, where the goal is to learn representations that are conditionally independent of a known distractor $Z$. We switch from $X$ to an *encoder* $\varphi_\theta(X)$. If the task is to predict $Y$ using some loss $L(\varphi_\theta(X), Y)$, the CIRCE regularized loss with the regularization weight $\gamma > 0$ is as follows:

$$\min_\theta L(\varphi_\theta(X), Y) + \gamma \, \mathrm{CIRCE}(\varphi_\theta(X), Z, Y) . \tag{12}$$

## 3 RELATED WORK

We review prior work on kernel-based measures of conditional independence to determine or enforce $X \perp\!\!\!\perp Z | Y$, including those measures we compare against in our experiments in Section 4. We begin with procedures based on kernel conditional feature covariances. The conditional kernel cross-covariance was first introduced as a measure of conditional dependence by Sun et al. (2007). Following this work, a kernel-based conditional independence test (KCI) was proposed by Zhang et al. (2011). The latter test relies on satisfying Proposition 2.2 leading to a statistic[4] that requires regression of $\varphi(X)$ on $Y$ in every minibatch (as well as of $Z$ on $Y$, as in our setting). More

---

[4]The conditional-independence test statistic used by KCI is $\frac{1}{B} \mathrm{Tr} \left( \tilde{K}_{\ddot{X}|Y} \tilde{K}_{Z|Y} \right)$, where $\ddot{X} = (X, Y)$ and $\tilde{K}$ is a centered kernel matrix. Unlike CIRCE, $\tilde{K}_{\ddot{X}|Y}$ requires regressing $\ddot{X}$ on $Y$ using kernel ridge regression.

recently, Quinzan et al. (2022) introduced a variant of the Hilbert-Schmidt Conditional Independence Criterion (HSCIC; Park & Muandet, 2020) as a regularizer to learn a generalized notion of counterfactually-invariant representations (Veitch et al., 2021). Estimating $\text{HSCIC}(X, Z|Y)$ from finite samples requires estimating the conditional mean-embeddings $\mu_{X,Z|Y}$, $\mu_{X|Y}$ and $\mu_{Z|Y}$ via regressions (Grunewalder et al., 2012). HSCIC requires three times as many regressions as CIRCE, of which two must be done online in minibatches to account for the conditional cross-covariance terms involving $X$. We will compare against HSCIC in experiements, being representative of this class of methods, and having been employed successfully in a setting similar to ours.

Alternative measures of conditional independence make use of additional normalization over the measures described above. The Hilbert-Schmidt norm of the *normalized* cross-covariance was introduced as a test statistic for conditional independence by Fukumizu et al. (2008), and was used for structure identification in directed graphical models. Huang et al. (2022) proposed using the ratio of the *maximum mean discrepancy* (MMD) between $P_{X|ZY}$ and $P_{X|Y}$, and the MMD between the Dirac measure at $X$ and $P_{X|Y}$, as a measure of the conditional dependence between $X$ and $Z$ given $Y$. The additional normalization terms in these statistics can result in favourable asymptotic properties when used in statistical testing. This comes at the cost of increased computational complexity, and reduced numerical stability when used as regularizers on minibatches.

Another approach, due to Shah & Peters (2020), is the Generalized Covariance Measure (GCM). This is a normalized version of the covariance between residuals from kernel-ridge regressions of $X$ on $Y$ and $Z$ on $Y$ (in the multivariate case, a maximum over covariances between univariate regressions is taken). As with the approaches discussed above, the GCM also involves multiple regressions – one of which (regressing $X$ on $Y$) cannot be done offline. Since the regressions are univariate, and since GCM simply regresses $Z$ and $X$ on $Y$ (instead of $\psi(Z, Y)$ and $\phi(X)$ on $Y$), we anticipate that GCM might provide better regularization than HSCIC on minibatches. This comes at a cost, however, since by using regression residuals rather than conditionally centered features, there will be instances of conditional dependence that will not be detectable. We will investigate this further in our experiments.

## 4 EXPERIMENTS

We conduct experiments addressing two settings: (1) synthetic data of moderate dimension, to study effectiveness of CIRCE at enforcing conditional independence under established settings (as envisaged for instance in econometrics or epidemiology); and (2) high dimensional image data, with the goal of learning image representations that are robust to domain shifts. We compare performance over all experiments with HSCIC (Quinzan et al., 2022) and GCM (Shah & Peters, 2020).

### 4.1 SYNTHETIC DATA

We first evaluate performance on the synthetic datasets proposed by Quinzan et al. (2022): these use the structural causal model (SCM) shown in Figure 1, and comprise 2 univariate and 2 multivariate cases (see Appendix E for details). Given samples of $A$, $Y$ and $Z$, the goal is to learn a predictor $\hat{B} = \varphi(A, Y, Z)$ that is counterfactually invariate to $Z$. Achieving this requires enforcing conditional independence $\varphi(A, Y, Z) \perp\!\!\!\perp Z|Y$. For all experiments on synthetic data, we used a fully connected network with 9 hidden layers. The inputs of the network were $A$, $Y$ and $Z$. The task is to predict $B$ and

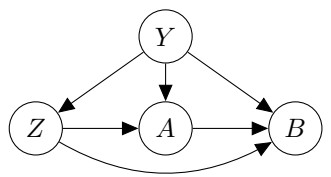

Figure 1: Causal structure for synthetic datasets.

the network is learned with the MSE loss. For each test case, we generated 10k examples, where 8k were used for training and 2k for evaluation. Data were normalized with zero mean and unit standard deviation. The rest of experimental details is provided in Appendix E.

We report in-domain MSE loss, and measure the level of counterfactual invariance of the predictor using the VCF (Quinzan et al., 2022, eq. 4; lower is better). Given $X = (A, Y, Z)$,

$$\text{VCF} := \mathbb{E}_{x \sim \mathbf{X}} \left[ \mathbb{V}_{z' \sim \mathbf{Z}} \left[ \mathbb{E}_{\hat{B}^*_{z'}|X} \left[ \hat{B}|X = x \right] \right] \right] . \tag{13}$$

$P_{\hat{B}^*_{z'}|X}$ is the counterfactual distribution of $\hat{B}$ given $X = x$ and an intervention of setting $z$ to $z'$.

**Univariate Cases**   Table 1 summarizes the in-domain MSE loss and VCF comparing CIRCE to baselines. Without regularization, MSE loss is low in-domain but the representation is not invariant to changes of $Z$. With regularization, all three methods successfully achieve counterfactual invariance in these simple settings, and exhibit similar in-domain performance.

| Case | No Reg | | GCM | | HSCIC | | CIRCE | |
|------|--------|-----|-----|-----|-------|-----|-------|-----|
| | MSE | VCF | MSE | VCF | MSE | VCF | MSE | VCF |
| 1 | 2.03e-4 | 0.180 | 0.198 | 2.59e-06 | 0.197 | 2.08e-11 | 0.197 | 8.77e-08 |
| 2 | 0.027 | 0.258 | 1.169 | 9.07e-07 | 1.168 | 3.08e-11 | 1.168 | 7.37e-11 |

Table 1: MSE loss and VCF for univariate synthetic datasets. Comparison of representation without conditional independence regularization against regularization with GCM, HSCIC and CIRCE.

**Multivariate Cases**   We present results on 2 multivariate cases: case 1 has high dimensional $Z$ and case 2 has high dimensional $Y$. For each multivariate case, we vary the number of dimensions $d = \{2, 5, 10, 20\}$. To visualize the trade-offs between in-domain performance and invariant representation, we plot the Pareto front of MSE loss and VCF. With high dimensional $Z$ (Figure 2A), CIRCE and HSCIC have a similar trade-off profile, however it is notable that GCM needs to sacrifice more in-domain performance to achieve the same level of invariance. This may be because the GCM statistic is a maximum over normalized covariances of univariate residuals, which can be less effective in a multivariate setting. For high dimensional $Y$ (Figure 2B), the regression from $Y$ to $\psi(Z)$ is much harder. We observe that HSCIC becomes less efficient with increasing $d$ until at $d = 20$ it fails completely, while GCM still sacrifices more in-domain performance than CIRCE.

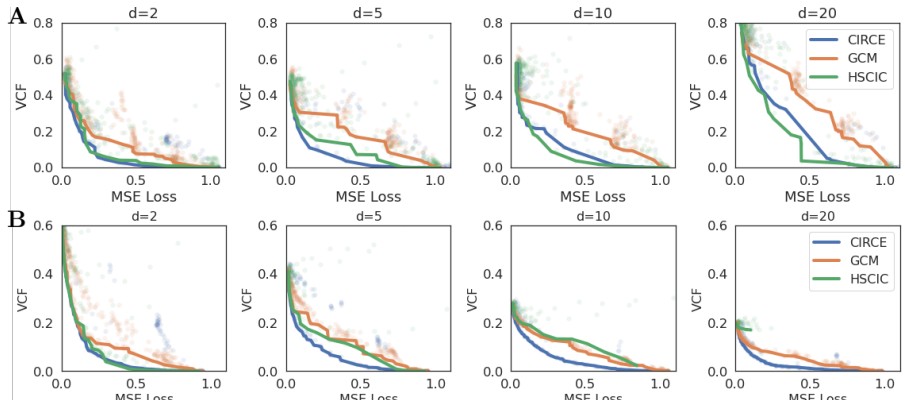

Figure 2: Pareto front of MSE and VCF for multivariate synthetic dataset. **A**: case 1; **B**: case 2.

## 4.2   IMAGE DATA

We next evaluate our method on two high-dimensional image datasets: d-Sprites (Matthey et al. (2017)) which contains images of 2D shapes generated from six independent latent factors; and the Extended Yale-B Face dataset [5](Georghiades et al. (2001)) of faces of 28 individuals under varying camera poses and illumination. We use both datasets with the causal graph in Figure 3 where the image $X$ is directly caused by the target variable $Y$ and a distractor $Z$. There also exists a strong non-causal association between $Y$ and $Z$ in the training set (denoted by the dashed edge).

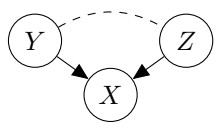

Figure 3: Causal structure for dSprites and Yale-B. Dashed line denotes a non-causal association between nodes.

The basic setting is as follows: for the in-domain (train) samples, the observed $Y$ and $Z$ are correlated through the true $Y$ as

$$Y \sim P_Y, \, \xi_z \sim \mathcal{N}(0, \sigma_z), \qquad Z = \beta(Y) + \xi_z, \tag{14}$$

$$Y' = Y + \xi_y, \, \xi_y \sim \mathcal{N}(0, \sigma_y), \quad Z' = f_z(Y, Z, \xi_z), \quad X = f_x(Y', Z'). \tag{15}$$

---

[5]Google and DeepMind do not have access or handle the Yale-B Face dataset.

$Y$ and $Z$ are observed; $f_z$ is the structural equation for $Z'$ (in the simplest case $Z' = Z$); $f_x$ is the generative process of $X$. $Y'$ and $Z'$ represent noise added during generation and are unobserved.

A regular predictor would take advantage of the association $\beta$ between $Z$ and $Y$ during training, since this is a less noisy source of information on $Y$. For unseen out-of-distribution (OOD) regime, where $Y$ and $Z$ are uncorrelated, such solution would be incorrect.

Therefore, our task is to learn a predictor $\hat{Y} = \varphi(X)$ that is conditionally independent of $Z$: $\varphi(X) \perp\!\!\!\perp Z \,|\, Y$, so that during the OOD/testing phase when the association between $Y$ and $Z$ ceases to exist, the model performance is not harmed as it would be if $\varphi(X)$ relied on the "shortcut" $Z$ to predict $Y$. For all image experiments we use the AdamW (Loshchilov & Hutter (2019)) optimizer and anneal the learning rate with a cosine scheduler (details in Appendix F). We select the hyper-parameters of the optimizer and scheduler via a grid search to minimize the in-domain validation set loss.

### 4.2.1 DSPRITES

Of the six independent generative factors in d-Sprites, we choose the $y$-coordinate of the object as our target $Y$ and the $x$-coordinate of the object in the image as our distractor variable $Z$. Our neural network consists of three convolutional layers interleaved with max pooling and leaky ReLU activations, followed by three fully-connected layers with 128, 64, 1 unit(s) respectively.

**Linear dependence** We sample images from the dataset as per the linear relation $Z' = Z = Y + \xi_z$. We then translate all sampled images (both in-domain and OOD) vertically by $\xi_y$, resulting in an observed object coordinate of $(Z, Y + \xi_y)$. In this case, linear residual methods, such as GCM, are able to sufficiently handle the dependence as the residual $Z - \mathbb{E}\left[Z \,|\, Y\right] = \xi_z$ is correlated with $Z$ – which is the observed $x$-coordinate. As a result, penalizing the cross-covariance between $\varphi(X) - \mathbb{E}\left[\varphi(X) \,|\, Y\right]$ and $Z - \mathbb{E}\left[Z \,|\, Y\right]$ will also penalize the network's dependence on the observed $x$-coordinate to predict $Y$.

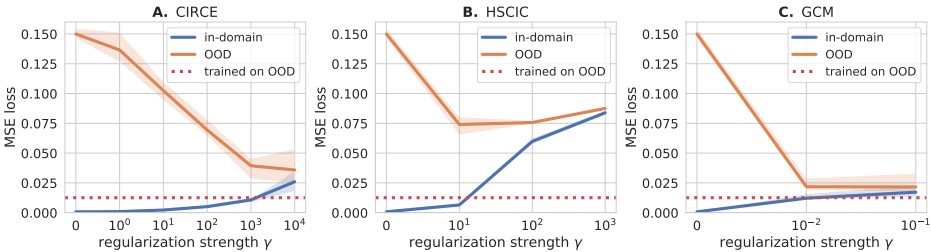

Figure 4: dSprites (linear). Blue: in-domain test loss; orange: out-of-domain loss (OOD); red: loss for OOD-trained encoder. Solid lines: median over 10 seeds; shaded areas: min/max values.

In Figure 4 we plot the in-domain and OOD losses over a range of regularization strengths and demonstrate that indeed GCM is able to perform quite well with a linear function relating $Z$ to $Y$. CIRCE is comparable to GCM with strong regularization and outperforms HSCIC. To get the optimal OOD baseline we train our network on an OOD training set where $Y$ and $Z$ are uncorrelated.

**Non-linear dependence** To demonstrate the limitation of GCM, which simply regresses $Z$ on $Y$ instead of $\psi(Z, Y)$ on $Y$, we next address a more complex nonlinear dependence $\beta(Y) = 0$ and $Z' = Y + \alpha Z^2$. The observed coordinate of the object in the image is $(Y + \alpha \xi_z^2, Y + \xi_y)$. For a small $\alpha$, the unregularized network will again exploit the shortcut, i.e. the observed $x$-coordinate, in order to predict $Y$. The linear residual, if we don't use features of $Z$, is $Z - \mathbb{E}\left[Z \,|\, Y\right] = \xi_z$, which is uncorrelated with $Y + \alpha \xi_z^2$, because $\mathbb{E}\left[\xi_z^3\right] = 0$ due to the symmetric and zero-mean distribution of $\xi_z$. As a result, penalizing cross-covariance with the linear residual (as done by GCM) will not penalize solutions that use the observed $x$-coordinate to predict $Y$. Whereas CIRCE which uses a feature map $\psi(Z)$ can capture higher order features. Results are shown in Figure 5: we see again that CIRCE performs best, followed by HSCIC, with GCM doing poorly. Curiously, GCM performance does still improve slightly on OOD data as regularization increases - we conjecture that the encoder $\varphi(X)$ may extract non-linear features of the coordinates. However, GCM is numerical unstable for large regularization weights, which might arise from combining a ratio normalization and a max operation in the statistic.

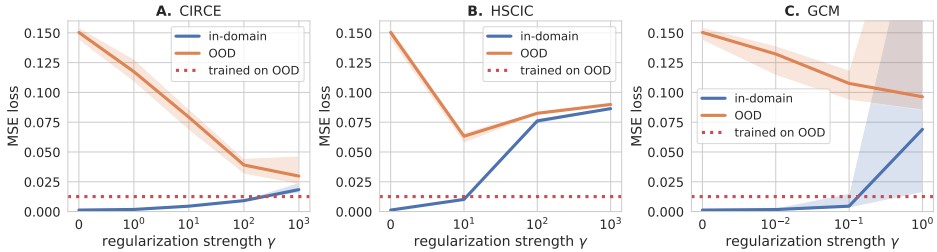

Figure 5: dSprites (non-linear). Blue: in-domain test loss; orange: out-of-domain loss (OOD); red: loss for OOD-trained encoder. Solid lines: median over 10 seeds; shaded areas: min/max values.

### 4.2.2 EXTENDED YALE-B

Finally, we evaluate CIRCE as a regressor for supervised tasks on the natural image dataset of Extended Yale-B Faces. The task here is to estimate the camera pose $Y$ from image $X$ while being conditionally independent of the illumination $Z$ which is represented as the azimuth angle of the light source with respect to the subject. Since, these are natural images, we use the ResNet-18 (He et al., 2016) model pre-trained on ImageNet (Deng et al., 2009) to extract image features, followed by three fully-connected layers containing 128, 64 and 1 unit(s) respectively. Here we sample the training data according to the non-linear relation $Z' = Z = 0.5(Y + \varepsilon Y^2)$, where $\varepsilon$ is either $+1$ or $-1$ with equal probability. In this case $\mathbb{E}[Z \mid Y] = 0.5Y + 0.5Y^2 \mathbb{E}[\varepsilon \mid Y] = 0.5Y$, and thus the linear residuals depend on $Y$. (In experiments, $Y$ and $\varepsilon$ are re-scaled to be in the same range. We avoid it here for simplicity.) Note that GCM can in principle find the correct solution using a linear decoder. Results are shown in Figure 6. CIRCE shows a small advantage over HSCIC in OOD performance for the best regularizer choice. GCM suffers from numerical instability in this example, which leads to poor performance.

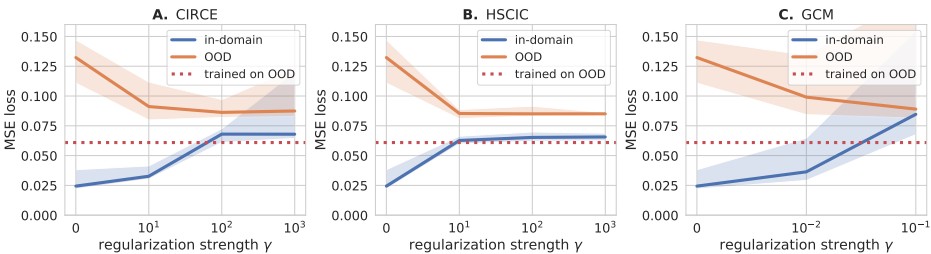

Figure 6: Yale-B. Blue: in-domain test loss; orange: out-of-domain loss (OOD); red: loss for OOD-trained encoder. Solid lines: median over 10 seeds; shaded areas: min/max values.

## 5 DISCUSSION

We have introduced CIRCE: a kernel-based measure of conditional independence, which can be used as a regularizer to enforce conditional independence between a network's predictions and a pre-specified variable with respect to which invariance is desired. The technique can be used in many applications, including fairness, domain invariant learning, and causal representation learning. Following an initial regression step (which can be done offline), CIRCE enforces conditional independence via a marginal independence requirement during representation learning, which makes it well suited to minibatch training. By contrast, alternative conditional independence regularizers require an additional regression step on each minibatch, resulting in a higher variance criterion which can be less effective in complex learning tasks.

As future work, it will be of interest to determine whether or not CIRCE is statistically significant on a given dataset, so as to employ it as a statistic for a test of conditional dependence.

ACKNOWLEDGMENTS

This work was supported by DeepMind, the Gatsby Charitable Foundation, the Wellcome Trust, the Canada CIFAR AI Chairs program, the Natural Sciences and Engineering Resource Council of Canada, SHARCNET, Calcul Québec, the Digital Resource Alliance of Canada, and Open Philanthropy. Finally, we thank Alexandre Drouin and Denis Therien for the Bellairs Causality workshop which sparked the project.

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

APPENDICES

## A  CONDITIONAL INDEPENDENCE DEFINITIONS

We first repeat the proof of the main theorem in Daudin 1980, as the missing proofs we need for the alternative definitions of independence rely on the main one.

**Theorem A.1** (Theorem 1 of Daudin 1980). *Define* $E_1 = \{g : g \in L^2_{XY}, \mathbb{E}[g \,|\, Y] = 0\}$, $E_2 = \{h : h \in L^2_{YZ}, \mathbb{E}[h \,|\, Y] = 0\}$. *Then, the following two conditions are equivalent:*

$$\mathbb{E}[g_1 h_1] = 0 \qquad\qquad \forall g_1 \in E_1, \forall h_1 \in E_2,$$
$$\mathbb{E}[gh \,|\, Y] = \mathbb{E}[g \,|\, Y]\,\mathbb{E}[h \,|\, Y] \qquad\qquad \forall g \in L^2_{XY}, \forall h \in L^2_{YZ}.$$

*Proof. Necessary condition:* $\mathbb{E}[gh \,|\, Y] = \mathbb{E}[g \,|\, Y]\,\mathbb{E}[h \,|\, Y] \implies \mathbb{E}[g_1 h_1] = 0$

Because $E_1 \subseteq L^2_{XY}$ and $E_2 \subseteq L^2_{YZ}$, for $g_1 \in E_1$ and $h_1 \in E_2$ we have

$$\mathbb{E}[g_1 h_1 \,|\, Y] = \mathbb{E}[g_1 \,|\, Y]\,\mathbb{E}[h_1 \,|\, Y] = 0$$
$$\implies \mathbb{E}[g_1 h_1] = \mathbb{E}_Y[\mathbb{E}[g_1 h_1 \,|\, Y]] = 0.$$

*Sufficient condition:* $\mathbb{E}[g_1 h_1] = 0 \implies \mathbb{E}[gh \,|\, Y] = \mathbb{E}[g \,|\, Y]\,\mathbb{E}[h \,|\, Y]$

Let $g' = g - \mathbb{E}[g \,|\, Y]$ where $g \in L^2_{XY}$ and $h' = h - \mathbb{E}[h \,|\, Y]$ where $h \in L^2_{XY}$. Then, $g' \in E_1$ and $h' \in E_2$

$$\begin{aligned}
\mathbb{E}[g'h'] &= \mathbb{E}[(g - \mathbb{E}[g \,|\, Y])(h - \mathbb{E}[h \,|\, Y])] \\
&= \mathbb{E}[gh - h\,\mathbb{E}[g \,|\, Y] - g\,\mathbb{E}[h \,|\, Y] + \mathbb{E}[g \,|\, Y]\,\mathbb{E}[h \,|\, Y]] \\
&= \mathbb{E}_Y[\mathbb{E}[(gh - h\,\mathbb{E}[g \,|\, Y] - g\,\mathbb{E}[h \,|\, Y] + \mathbb{E}[g \,|\, Y]\,\mathbb{E}[h \,|\, Y]) \,|\, Y]] \\
&= \mathbb{E}_Y[\mathbb{E}[gh \,|\, Y] - \mathbb{E}[g \,|\, Y]\,\mathbb{E}[h \,|\, Y]] = 0.
\end{aligned} \tag{16}$$

Let $B$ be a Borel set of the image space of $Y$, $g* = gI_B$ where $I_B$ is an indicator function of $B$. We have $\int g^{*2} dP = \int g^2 I_B dP = \int_B g^2 dP \le \int g^2 dP < \infty$, therefore $g^* \in L^2_{XY}$. Using Equation (16),

$$\begin{aligned}
&\mathbb{E}_Y[\mathbb{E}[g^* h \,|\, Y] - \mathbb{E}[g^* \,|\, Y]\,\mathbb{E}[h \,|\, Y]] \\
&= \mathbb{E}_Y[\mathbb{E}[ghI_B \,|\, Y] - \mathbb{E}[gI_B \,|\, Y]\,\mathbb{E}[h \,|\, Y]] \\
&= \int_B \mathbb{E}[gh \,|\, Y]\, dP - \int_B \mathbb{E}[g \,|\, Y]\,\mathbb{E}[h \,|\, Y]\, dP = 0
\end{aligned}$$

So $\mathbb{E}[gh \,|\, Y] = \mathbb{E}[g \,|\, Y]\,\mathbb{E}[h \,|\, Y]$ almost surely. $\qquad\square$

**Corollary A.2** (Equation 3.8 of Daudin 1980). *The following two conditions are equivalent:*

$$\mathbb{E}[gh_1] = 0 \qquad\qquad \forall g \in L^2_{XY}, \forall h_1 \in E_2,$$
$$\mathbb{E}[gh \,|\, Y] = \mathbb{E}[g \,|\, Y]\,\mathbb{E}[h \,|\, Y] \qquad\qquad \forall g \in L^2_{XY}, \forall h \in L^2_{YZ}.$$

*Proof. Necessary condition* is identical to the previous proof.

*Sufficient condition:* $\mathbb{E}[gh_1] = 0 \implies \mathbb{E}[gh \,|\, Y] = \mathbb{E}[g \,|\, Y]\,\mathbb{E}[h \,|\, Y]$

Let $h' = h - \mathbb{E}[h \,|\, Y]$ where $h \in L^2_{YZ}$, then $h' \in E_2$

$$\begin{aligned}
\mathbb{E}[gh'] &= \mathbb{E}[g(h - \mathbb{E}[h \,|\, Y])] \\
&= \mathbb{E}[gh - g\,\mathbb{E}[h \,|\, Y]] \\
&= \mathbb{E}_Y[\mathbb{E}[(gh - g\,\mathbb{E}[h \,|\, Y]) \,|\, Y]] \\
&= \mathbb{E}_Y[\mathbb{E}[gh \,|\, Y] - \mathbb{E}[g\,\mathbb{E}[h \,|\, Y] \,|\, Y]] \\
&= \mathbb{E}_Y[\mathbb{E}[gh \,|\, Y] - \mathbb{E}[g \,|\, Y]\,\mathbb{E}[h \,|\, Y]] = 0.
\end{aligned}$$

Using the same argument as for Theorem A.1, $\mathbb{E}[gh \,|\, Y] = \mathbb{E}[g \,|\, Y]\,\mathbb{E}[h \,|\, Y]$ almost surely. $\qquad\square$

**Corollary A.3** (Equation 3.9 of Daudin 1980). *The following two conditions are equivalent:*

$$\mathbb{E}\left[g'h_1\right] = 0 \qquad\qquad \forall g' \in L_X^2, \forall h_1 \in E_2\,,$$
$$\mathbb{E}\left[gh\,|\,Y\right] = \mathbb{E}\left[g\,|\,Y\right]\mathbb{E}\left[h\,|\,Y\right] \qquad\qquad \forall g \in L_{XY}^2, \forall h \in L_{YZ}^2\,.$$

*Proof. Necessary condition:* As $E_2 \subseteq L_{YZ}^2$ and $L_X^2 \subseteq L_{XY}^2$,

$$\mathbb{E}\left[g'h_1\,|\,Y\right] = \mathbb{E}\left[g'\,|\,Y\right]\mathbb{E}\left[h_1\,|\,Y\right] = 0\,.$$

*Sufficient condition:* $\mathbb{E}[g'h_1] = 0 \implies \mathbb{E}\left[gh\,|\,Y\right] = \mathbb{E}\left[g\,|\,Y\right]\mathbb{E}\left[h\,|\,Y\right]$

Take a simple function $g_a = \sum_{i=1}^n a_i I_{A_i}$ for an integrable Borel set $A_i$ in $XY$. As integrable simple functions are dense in $L_{XY}^2$, we only need to prove the condition for all $g_a$.

In our case, the indicator function decomposes as $I_{A_i} = I_{A_i^X} I_{A_i^Y}$, and therefore for $g_i = a_i I_{A_i^X}$

$$g_a = \sum_i^n g_i I_{A_i^Y}\,.$$

Therefore,

$$\mathbb{E}[g_a h_1] = \mathbb{E}\left[\sum_{i=1}^n I_{A_i^Y}\,\mathbb{E}\left[g_i h_1\,|\,Y\right]\right] = \mathbb{E}\left[\sum_{i=1}^n I_{A_i^Y}\cdot 0\right] = 0\,.$$

As simple functions are dense in $L_{XY}^2$, we immediately have $\mathbb{E}[gh_1] = 0\ \forall g \in L_{XY}^2, h_1 \in E_2$. Applying Corollary A.2 concludes the proof.

$\square$

# B CIRCE DEFINITION

First, we need a more convenient function class:

**Lemma B.1.** *The function class* $E_2 = \left\{h \in L_{ZY}^2, \mathbb{E}\left[h\,|\,Y\right] = 0\right\}$ *coincides with the function class* $E_2' = \left\{h' = h - \mathbb{E}\left[h\,|\,Y\right],\ h \in L_{ZY}^2\right\}$.

*Proof.* $E_2 \subseteq E_2'$: any $h \in E_2$ is in $L_{ZY}^2$ and has the form $h = h - \mathbb{E}\left[h\,|\,Y\right]$ by construction because the last term is zero.

$E_2' \subseteq E_2$: first, any $h' \in E_2'$ satisfies $\mathbb{E}\left[h'\,|\,Y\right] = 0$ by construction. Second,

$$\int (h')^2\,d\mu(Z,Y) = \int (h - \mathbb{E}\left[h\,|\,Y\right])^2\,d\mu(Z,Y) \tag{17}$$

$$= \int \left(h^2 - 2h\,\mathbb{E}\left[h\,|\,Y\right] + (\mathbb{E}\left[h\,|\,Y\right])^2\right)\,d\mu(Z,Y) \tag{18}$$

$$= \int \left(h^2 - (\mathbb{E}\left[h\,|\,Y\right])^2\right)\,d\mu(Z,Y) < +\infty\,, \tag{19}$$

as $h \in L_{ZY}^2$ and the second term is non-positive.

$\square$

*Proof of Theorem 2.5.* For the "if" direction, we simply "pull out" the $Y$ expectation in the definition of the CIRCE operator and apply conditional independence:

$$C_{XZ|Y}^c = \mathbb{E}_Y\left[\mathbb{E}_X[\phi(X)\,|\,Y] \otimes \underbrace{\left(\mathbb{E}_Z[\psi(Z,Y)\,|\,Y] - \mathbb{E}_{Z'}[\psi(Z',Y)\,|\,Y]\right)}_{0}\right] = 0.$$

For the other direction, first, $\|C_{XQ}^c\|_{\mathrm{HS}} = 0$ implies that for any $g \in \mathcal{G}$ and $h \in \mathcal{F}$,

$$\mathbb{E}\left[g\left(h - \mathbb{E}\left[h\,|\,Y\right]\right)\right] = 0 \tag{20}$$

by Cauchy-Schwarz.

Now, we use that an $L_2$-universal kernel is dense in $L^2$ by definition (see Sriperumbudur et al. (2011)). Therefore, for any $g \in L_X^2$ and $h \in L_{ZY}^2$, for any $\epsilon > 0$ we can find $g_\epsilon \in \mathcal{G}$ and $h_\epsilon \in \mathcal{F}$ such that

$$\|g - g_\epsilon\|_2 \leq \epsilon, \ \|h - h_\epsilon\|_2 \leq \epsilon. \tag{21}$$

For the $L^2$ function, we can now write the conditional independence condition as

$$\mathbb{E}\left[g\left(h - \mathbb{E}\left[h \mid Y\right]\right)\right] = \mathbb{E}\left[\left(g \pm g_\epsilon\right)\left(h \pm h_\epsilon - \mathbb{E}\left[h \pm h_\epsilon \mid Y\right]\right)\right] \tag{22}$$

$$= 0 + \mathbb{E}\left[\left(g - g_\epsilon\right)\left(h - h_\epsilon - \mathbb{E}\left[h - h_\epsilon \mid Y\right]\right)\right] \tag{23}$$

$$+ \mathbb{E}\left[g_\epsilon\left(h - h_\epsilon - \mathbb{E}\left[h - h_\epsilon \mid Y\right]\right)\right] - \mathbb{E}\left[\left(g - g_\epsilon\right)\left(h_\epsilon - \mathbb{E}\left[h_\epsilon \mid Y\right]\right)\right]. \tag{24}$$

The first term is zero because $\|C_{XQ}^c\|_{\mathrm{HS}} = 0$. For the rest, we need to apply Cauchy-Schwarz:

$$\mathbb{E}\left[\left(g - g_\epsilon\right)\left(h - h_\epsilon\right)\right] \leq \|g - g_\epsilon\|_2 \|h - h_\epsilon\|_2 \leq \epsilon^2 \tag{25}$$

$$\mathbb{E}\left[\left(g - g_\epsilon\right)\left(\mathbb{E}\left[h - h_\epsilon \mid Y\right]\right)\right] \leq \|g - g_\epsilon\|_2 \|h - h_\epsilon\|_2 \leq \epsilon^2, \tag{26}$$

where in the last inequality we used that $\mathbb{E}\left[\left(\mathbb{E}\left[X \mid H\right]\right)^2\right] \leq \mathbb{E}\left[X^2\right]$ for conditional expectations.

Similarly, also using the reverse triangle inequality,

$$\mathbb{E}\left[g_\epsilon\left(h - h_\epsilon\right)\right] \leq \epsilon \|g_\epsilon\|_2 \leq \epsilon\left(\|g\|_2 + \epsilon\right). \tag{27}$$

Repeating this calculation for the rest of the terms, we can finally apply the triangle inequality to show that

$$\left|\mathbb{E}\left[g\left(h - \mathbb{E}\left[h \mid Y\right]\right)\right]\right| \leq 2\,\epsilon^2 + 2\,\epsilon\left(\|g\|_2 + \epsilon\right) + 2\,\epsilon\left(\|h\|_2 + \epsilon\right) \tag{28}$$

$$= 2\,\epsilon\left(3\,\epsilon + \|g\|_2 + \|h\|_2\right). \tag{29}$$

As $\|g\|_2$ and $\|h\|_2$ are fixed and finite, we can make the bound arbitrary small, and hence $\mathbb{E}\left[g\left(h - \mathbb{E}\left[h \mid Y\right]\right)\right] = 0$. □

## C  PROOFS FOR ESTIMATORS

### C.1  ESTIMATING THE CONDITIONAL MEAN EMBEDDING

We will construct an estimate of the term $\mathbb{E}_Z\left[\psi(Z, Y) \mid Y\right]$ that appears inside CIRCE, as a function of $Y$. We summarize the established results on conditional feature mean estimation: see (Grunewalder et al., 2012; Park & Muandet, 2020; Mollenhauer & Koltai, 2020; Klebanov et al., 2020; Li et al., 2022) for further details. To learn $\mathbb{E}\left[\psi(Q) \mid Y\right]$ for some feature map $\psi(q) \in \mathcal{H}_Q$ and random variable $Q$ (both to be specified shortly), we can minimize the following loss:

$$\hat{\mu}_{Q \mid Y, \lambda}(y) = \underset{F \in \mathcal{G}_{QY}}{\arg\min} \sum_{i=1}^{N} \|\psi(q_i) - F(y_i)\|_{\mathcal{H}_Q}^2 + \lambda \|F\|_{\mathcal{G}_{QY}}^2, \tag{30}$$

where $\mathcal{G}_{QY}$ is the space of functions from $Y$ to $\mathcal{H}_Q$. The above solution is said to be *well-specified* when there exists a Hilbert-Schmidt operator $A^* \in HS(\mathcal{H}_Y, \mathcal{H}_Q)$ such that $F^*(y) = A^*\psi(y)$ for all $y \in \mathcal{Y}$, where $\mathcal{H}_Y$ is the RKHS on $\mathcal{Y}$ with feature map $\psi(y)$ (Li et al., 2022).

We now consider the case relevant to our setting, where $Q := (Z, Y)$. We define[6] $\psi(Z, Y) = \psi(Z) \otimes \psi(Y)$, which for radial basis kernels (e.g. Gaussian, Laplace) is $L_2$-universal for $(Z, Y)$.[7] We then write $\mathbb{E}_Z\left[\psi(Z, y) \mid Y = y\right] = \mathbb{E}_Z\left[\psi(Z) \mid Y = y\right] \otimes \psi(y)$. The conditional feature mean $\mathbb{E}\left[\psi(Z) \mid Y\right]$ can be found with kernel ridge regression (Grunewalder et al., 2012; Li et al., 2022):

$$\mu_{Z \mid Y}(y) \equiv \mathbb{E}\left[\psi(Z) \mid Y\right](y) \approx K_{yY}\left(K_{YY} + \lambda I\right)^{-1} K_Z. \tag{31}$$

---

[6]We abuse notation in using $\psi$ to denote feature maps of $(Y, Z)$, $Y$, and $Z$; in other words, we use the argument of the feature map to specify the feature space, to simplify notation.

[7]Fukumizu et al. (2008, Section 2.2) show this kernel is *characteristic*, and Sriperumbudur et al. (2011, Figure 1 (3)) that being characteristic implies $L_2$ universality in this case.

where $K_Z$. indicates a "matrix" with rows $\psi(z_i)$, $(K_{YY})_{i,j} = k(y_i, y_j)$, and $(K_{yY})_i = k(y, y_i)$. Note that we have used the argument of $k$ to identify which feature space it pertains to – i.e., the kernel on $Z$ need not be the same as that on $Y$.

We can find good choices for the $Y$ kernel and the ridge parameter $\lambda$ by minimizing the leave-one-out cross-validation error. In kernel ridge regression, this is almost computationally free, based on the following version of a classic result for scalar-valued ridge regression. The proof generalizes the proof of Theorem 3.2 of Bachmann et al. (2022) to RKHS-valued outputs.

**Theorem C.1** (Leave-one-out for kernel mean embeddings). *Denote the predictor trained on the full dataset as $F_S$, and the one trained without the $i$-th point as $F_{-i}$. For $\lambda > 0$ and $A \equiv K_{YY} (K_{YY} + \lambda I)^{-1}$, the leave-one-out (LOO) error for Equation (30) is*

$$\frac{1}{N} \sum_{i=1}^{N} \|\psi(z_i) - F_{-i}(y_i)\|_{\mathcal{H}_Z}^2 = \frac{1}{N} \sum_{i=1}^{N} \frac{\|\psi(z_i) - F_S(y_i)\|_{\mathcal{H}_Z}^2}{(1 - A_{ii})^2} . \tag{32}$$

*Proof.* Denote the full dataset $S = \{(y_i, z_i)\}_{i=1}^{M}$; the dataset missing the $i$-th point is denoted $S_{-i}$. Prediction on the full dataset takes the form $F(Y) = A K_Z$ ..

Consider the prediction obtained without the $M$-th point (w.l.o.g.) but evaluated on $y_M$: $F_{-M}(y_M)$. Define a new dataset $\mathcal{Z} = S_{-M} \cup \{(y_M, F_{-M}(y_M))\}$ and compute the loss for it:

$$L_{\mathcal{Z}}(F_{-M}) = \sum_{i=1}^{M-1} \|\psi(z_i) - F_{-M}(y_i)\|_{\mathcal{H}_Z}^2 + \|F_{-M}(y_M) - F_{-M}(y_M)\|_{\mathcal{H}_Z}^2 + \lambda \|F_{-M}\|_{\mathcal{G}_{ZY}}^2 \tag{33}$$

$$= L_{S_{-M}}(F_{-M}) \leq L_{S_{-M}}(F) \leq L_{S_{-M}}(F) + \|F_{-M}(y_M) - F(y_M)\|_{\mathcal{H}_Z}^2 \leq L_{\mathcal{Z}}(F), \tag{34}$$

where the first inequality is due to $F_{-M}$ minimizing $L_{S_{-M}}$. Therefore, $F_{-M}$ also minimizes $L_{\mathcal{Z}}$. As $A$ in the prediction expression $F_S(Y) = A K_Z$. depends only on $Y$, and not on $Z$, $F_{-M}$ has to have the same form as the full prediction:

$$F_{-M}(Y) = A K_{\tilde{Z}} ., \quad K_{\tilde{z}_i, .} = \begin{cases} \psi(z_i), & i < M, \\ F_{-M}(y_M), & i = M. \end{cases} \tag{35}$$

This allows us to solve for $F_{-M}(y_M)$:

$$F_{-M}(y_M) = K_{y_M Y} (K_{YY} + \lambda I)^{-1} K_{\tilde{Z}} . = \sum_{i=1}^{M} A_{Mi} \psi(z_i) \tag{36}$$

$$= \sum_{i=1}^{M-1} A_{Mi} \psi(z_i) + A_{MM} \psi(z_i) \pm A_{MM} \psi(z_M) \tag{37}$$

$$= \sum_{i=1}^{M} A_{Mi} \psi(z_i) - A_{MM} \psi(z_M) + A_{MM} \psi(z_i) \tag{38}$$

$$= F_S(y_M) - A_{MM} \psi(z_M) + A_{MM} \psi(z_i) \tag{39}$$

$$= F_S(y_M) - A_{MM} \psi(z_M) + A_{MM} F_{-M}(y_M) . \tag{40}$$

As $A_{MM}$ is a scalar, we can solve for $F_{-M}(y_M)$:

$$F_{-M}(y_M) = \frac{F_S(y_M) - A_{MM} \psi(z_M)}{1 - A_{MM}} \tag{41}$$

Therefore,

$$\psi(z_M) - F_{-M}(y_M) = \frac{(1 - A_{MM}) \psi(z_M) - F_S(y_M) + A_{MM} \psi(z_M)}{1 - A_{MM}} \tag{42}$$

$$= \frac{\psi(z_M) - F_S(y_M)}{1 - A_{MM}} . \tag{43}$$

Taking the norm and summing this result over all points (not just $M$) gives the LOO error. $\qquad \square$

## C.2 CIRCE ESTIMATORS

**Lemma C.2.** *For B points and $K_{zz'}^c = \langle \psi(z) - \mathbb{E}\left[Z \,|\, Y\right](y), \psi(z') - \mathbb{E}\left[Z \,|\, Y\right](y')\rangle$, the CIRCE estimator*

$$\|\widehat{C_{XZ|Y}^c}\|^2_{\mathrm{HS}} = \frac{1}{B(B-1)}\mathrm{Tr}\left(K_{XX}(K_{YY} \odot K_{ZZ}^c)\right) \tag{44}$$

*has $O(1/B)$ bias and $O_p(1/\sqrt{B})$ deviation from the mean for any fixed probability of the deviation.*

*Proof.* The bias is straightforward:

$$\frac{1}{B(B-1)}\mathbb{E}\left[\mathrm{Tr}\left(K_{XX}(K_{YY} \odot K_{ZZ}^c)\right)\right]$$

$$= \frac{1}{B(B-1)}\mathbb{E}\left[\sum_{i,j \neq i} K_{x_i x_j} K_{y_i y_j} K_{z_i z_j}^c\right] + \frac{1}{B(B-1)}\mathbb{E}\left[\sum_i K_{x_i x_i} K_{y_i y_i} K_{z_i z_i}^c\right]$$

$$= \frac{1}{B(B-1)}\sum_{i,j \neq i}\mathbb{E}_{xx'yy'zz'}\left[K_{xx'} K_{yy'} K_{zz'}^c\right] + O\left(\frac{1}{B}\right)$$

$$= \|C_{XQ}^c\|^2_{\mathrm{HS}} + O\left(\frac{1}{B}\right).$$

For the variance, first note that our estimator has bounded differences. Denote $K_{TT} = K_{YY} \odot K_{ZZ}^c$ and $t = (y, z)$, if we switch one datapoint $(x_i, t_i)$ to $(x_i', t_i')$ and denote the vectors with switch coordinates as $X^i, T^i$

$$\left|\mathrm{Tr}\left(K_{XX} K_{TT}\right) - \mathrm{Tr}\left(K_{X^i X^i} K_{T^i T^i}\right)\right|$$

$$= \left| K_{x_i x_i} K_{t_i t_i} - K_{x_i' x_i'} K_{t_i' t_i'} + 2\sum_{j \neq i}\left(K_{x_j x_i} K_{t_j t_i} - K_{x_j x_i'} K_{t_j t_i'}\right)\right|$$

$$\leq (2 + 4(B-1))K_{x\,\max}K_{t\,\max} \leq (4B - 2)K_{x\,\max}K_{y\,\max}K_{z\,\max}^c.$$

Therefore, for any index $i$

$$\frac{1}{B(B-1)}\left|\mathrm{Tr}\left(K_{XX}\left(K_{YY} \odot K_{ZZ}^c\right)\right) - \mathrm{Tr}\left(K_{X^i X^i}\left(K_{Y^i Y^i} \odot K_{Z^i Z^i}^c\right)\right)\right|$$

$$\leq \frac{4B - 2}{B(B-1)}K_{x\,\max}K_{y\,\max}K_{z\,\max}^c.$$

We can now use McDiarmid's inequality (McDiarmid, 1989) with

$$c = c_i = \frac{4B - 2}{B(B-1)}K_{x\,\max}K_{y\,\max}K_{z\,\max}^c,$$

meaning that for any $\epsilon > 0$

$$\mathrm{P}\left(\left|\frac{\mathrm{Tr}\left(K_{XX} K_{TT}\right)}{B(B-1)} - \mathbb{E}\,\frac{\mathrm{Tr}\left(K_{XX} K_{TT}\right)}{B(B-1)}\right| \geq \epsilon\right) \leq 2\exp\left(-\frac{2\epsilon^2}{Bc^2}\right)$$

$$= 2\exp\left(-\frac{2\epsilon^2 B(B-1)^2}{(4B-2)^2 K_{x\,\max}^2 K_{y\,\max}^2 K_{z\,\max}^{2c}}\right).$$

Therefore, for any fixed probability the deviation $\epsilon$ from the mean decays as $O(1/\sqrt{B})$. □

**Definition C.3.** A $(\beta, p)$-kernel for a given data distribution satisfies the following conditions (see Fischer & Steinwart (2020); Li et al. (2022) for precise definition using interpolation spaces):

(EVD) Eigenvalues $\mu_i$ of the covariance operator $C_{YY}$ decay as $\mu_i \leq c \cdot i^{-1/p}$.

(EMB) For $\alpha \in (p, 1]$, the inclusion map $[\mathcal{H}_Y^\alpha \hookrightarrow L_\infty(\pi)]$ is continuous and bounded by $A$.

(SRC) $F \in [\mathcal{G}]^{\beta}$ for $\beta \in [1, 2]$ (note that $\beta < 1$ would include the misspecified setting).

**Lemma C.4.** *Consider the well-specified case of conditional expectation estimation (see Li et al., 2022). For bounded kernels over $X, Z, Y$ and a $(\beta, p)$-kernel over $Y$, $F(y) = \mathbb{E}\left[\psi(Z) \mid Y\right](y)$, bounded $\|F\| \leq C_F$, and $M$ points used to estimate $F$, define the conditional expectation estimate as*

$$\hat{F}(y) = K_{yY} \left(K_{YY} + \lambda_M I\right)^{-1} K_{Z\,\cdot}\,, \tag{45}$$

*where $\lambda_M = \Theta(1/M^{\beta+p})$.*

*Then, the estimator $\mathrm{Tr}\left(K_{XX} \hat{K}_{ZZ}^c\right) / (B(B-1))$ of the "true" CIRCE estimator (i.e., with the actual conditional expectation) deviates from the true value as $O_p(1/M^{(\beta-1)/(2(\beta+p))})$.*

*Proof.* First, decompose the difference:

$$\mathrm{Tr}\left(K_{XX} K_{ZZ}^c\right) - \mathrm{Tr}\left(K_{XX} \hat{K}_{ZZ}^c\right) = \mathrm{Tr}\left(K_{XX} \left(K_{ZZ}^c - K_{ZZ}^c\right)\right) \tag{46}$$

$$= \mathrm{Tr}\left(K_{XX}\left[\left(K_{ZZ}^c - \hat{K}_{ZZ}^c\right) \odot K_{YY}\right]\right) = \mathrm{Tr}\left(\left[K_{XX} \odot K_{YY}\right]\left(K_{ZZ}^c - \hat{K}_{ZZ}^c\right)\right), \tag{47}$$

where in the last line we used that all matrices are symmetric.

Let's concentrate on the difference:

$$\left(K_{ZZ}^c - \hat{K}_{ZZ}^c\right)_{ij} = \left\langle \hat{F}(y_i) - F(y_i),\, \psi(z_j)\right\rangle + \left\langle \hat{F}(y_j) - F(y_j),\, \psi(z_i)\right\rangle \tag{48}$$

$$+ \left\langle F(y_i),\, F(y_j)\right\rangle - \left\langle \hat{F}(y_i),\, \hat{F}(y_j) \pm F(y_j)\right\rangle \tag{49}$$

$$= \left\langle \hat{F}(y_i) - F(y_i),\, \psi(z_j)\right\rangle + \left\langle \hat{F}(y_j) - F(y_j),\, \psi(z_i)\right\rangle \tag{50}$$

$$+ \left\langle F(y_i) - \hat{F}(y_i),\, F(y_j)\right\rangle - \left\langle \hat{F}(y_i),\, \hat{F}(y_j) - F(y_j)\right\rangle \tag{51}$$

$$= \left\langle F(y_i) - \hat{F}(y_i),\, F(y_j) - \psi(z_j)\right\rangle + \left\langle F(y_j) - \hat{F}(y_j),\, \hat{F}(y_i) - \psi(z_j)\right\rangle. \tag{52}$$

As we're working in the well-specified case, by definition the operator $F \in \mathcal{G}$, where $\mathcal{G}$ is a vector-valued RKHS (Li et al., 2022, Definition 1). This implies that for the function $[K_x h](\cdot) = K(\cdot, x)h$ (where $h \in \mathcal{H}_y$),

$$\langle F(x),\, h\rangle = \langle F,\, K_x h\rangle_{\mathcal{G}}\,. \tag{53}$$

We can now re-write the difference as

$$\left(K_{ZZ}^c - \hat{K}_{ZZ}^c\right)_{ij} = \left\langle F - \hat{F},\, K_{y_i}\left(F(y_j) - \psi(z_j)\right) + K_{y_j}\left(\hat{F}(y_i) - \psi(z_j)\right)\right\rangle_{\mathcal{G}}. \tag{54}$$

We can use the triangle inequality and then Cauchy-Schwarz to obtain

$$\left|\left(K_{ZZ}^c - \hat{K}_{ZZ}^c\right)_{ij}\right| \leq \|F - \hat{F}\|_{\mathcal{G}}\left(\left\|K_{y_i}\left(F(y_j) - \psi(z_j)\right)\right\|_{\mathcal{G}} + \left\|K_{y_j}\left(\hat{F}(y_i) - \psi(z_j)\right)\right\|_{\mathcal{G}}\right) \tag{55}$$

$$= \|F - \hat{F}\|_{\mathcal{G}}\left(k(y_i, y_i)\|F(y_j) - \psi(z_j)\|_{\mathcal{H}_Z} + k(y_j, y_j)\|\hat{F}(y_i) - \psi(z_j)\|_{\mathcal{H}_Z}\right) \tag{56}$$

$$\leq C_1 \|F - \hat{F}\|_{\mathcal{G}}\left(C_2 + C_3 \|F - \hat{F}\|_{\mathcal{G}}\right), \tag{57}$$

for some positive constants $C_{1,2,3}$ (since the kernels over both $z$ and $y$ are bounded, $F$ is bounded too and hence $\|\hat{F}\| \leq \|\hat{F} - F\| + \|F\|$.

As all kernels are bounded,

$$\frac{\left|\mathrm{Tr}\left(\left[K_{XX} \odot K_{YY}\right]\left(K_{ZZ}^c - \hat{K}_{ZZ}^c\right)\right)\right|}{B(B-1)} \leq C_1 C_4 \|F - \hat{F}\|_{\mathcal{G}}\left(C_2 + C_3 \|F - \hat{F}\|_{\mathcal{G}}\right) \tag{58}$$

for positive constants $C_1$ to $C_4$.

Now we can use Theorem 2 of Li et al. (2022) with $\gamma = 1$ and $\lambda = \Theta(1/M^{\beta+p})$, which shows that

$$\mathrm{P}\left( \|F - \hat{F}\|_{\mathcal{G}} \leq \tau \sqrt{K} M^{-\frac{\beta-1}{2(\beta+p)}} \right) \geq 1 - 4e^{-\tau}, \tag{59}$$

for some positive constant $K$, which gives us the $O_p(1/M^{\frac{\beta-1}{2(\beta+p)}})$ deviation. $\qquad \square$

Now we can combine the two lemmas to prove Theorem 2.7:

*Proof of Theorem 2.7.* Combining Lemma C.2 and Lemma C.4 and using a union bound, we obtain the $O_p(1/\sqrt{B} + 1/M^{\frac{\beta}{2(\beta+p)}})$ rate. $\qquad \square$

**Corollary C.5.** *For $B$ points and $M$ holdout points, the CIRCE estimator*

$$\widehat{CIRCE} = \frac{1}{B(B-1)} \mathrm{Tr}\left( \tilde{K}_{XX} \left( \tilde{K}_{YY} \odot \hat{\tilde{K}}_{ZZ}^c \right) \right), \quad \tilde{A} = A - \mathrm{diag}(A), \tag{60}$$

*converges as $O_p(1/\sqrt{B} + 1/M^{\frac{\beta-1}{2(\beta+p)}})$.*

*Proof.* This follows from the previous two proofs. $\qquad \square$

**Corollary C.6.** *For $B$ points and $M$ holdout points, the CIRCE estimator*

$$\widehat{CIRCE} = \frac{1}{B(B-1)} \mathrm{Tr}\left( H K_{XX} H \left( K_{YY} \odot \hat{K}_{ZZ}^c \right) \right), \quad H = I - \frac{1}{B} 1_B 1_B^\top \tag{61}$$

*has bias of $O(1/B)$ and converges as $O_p(1/\sqrt{B} + 1/M^{\frac{\beta-1}{2(\beta+p)}})$.*

*Proof.* This follows from the previous two proofs and the fact that $K^c$ is a centered matrix, meaning that in expectation $HK^cH = K^c$. $\qquad \square$

This estimator can be less biased in practice, as $\hat{K}_{ZZ}^c$ is typically biased due to conditional expectation estimation, and $H\hat{K}^cH$ re-centers it.

## D    RANDOM FOURIER FEATURES

Random Fourier features (RFF) Rahimi & Recht (2007) allow to approximate a kernel $k(x_1, x_2) \approx \frac{1}{D} \sum_{i=1}^D r_i(x_1)^\top r_i(x_2)$, and therefore $K = RR^\top$.

The algorithm to estimate CIRCE with RFF is provided in Algorithm 2. We sample $D_0$ points every $L$ iterations, but in every batch only use $D$ of them to reduce computational costs. It takes $O(D_0 M^2 + D_0^2 M)$ to compute $W_1^\tau$ and $W_2^\tau$ every $L$ iterations. At each iteration, it takes $O(BD^2 + B^2 D)$ to compute CIRCE. Therefore, average (per iteration) cost of RFF estimation becomes $O(\frac{D_0}{L} M^2 + \frac{D_0^2}{L} M + BD^2 + B^2 D)$.

## E    SYNTHETIC DATA AND ADDITIONAL RESULTS

We used Adam (Kingma & Ba, 2015) for optimization with batch size 256, and trained the network for 100 epochs. For experiments on univariate datasets, the learning rate was 1e-4 and weight decay was 0.3; for experiments on multivariate datasets, the learning rate was 3e-4 and weight decay was 0.1. We implemented CIRCE with random Fourier features (Rahimi & Recht, 2007) (see Appendix D) of dimension 512 for Gaussian kernels. We swept over the hyperparameters, including RBF scale, regularization weight for ridge regression, and regularization weight for the conditional independence regularization strength.

All synthetic datasets are using the same causal structure as shown in Figure 1. Hyperparameters sweep is listed in Table 2 and it is the same for all test cases.

---

**Algorithm 2** Estimation of CIRCE with random Fourier features

---

Holdout data $\{(z_i, y_i)\}_{i=1}^M$, mini-batch $\{(x_i, z_i, y_i)\}_{i=1}^B$
**Holdout data**
Leave-one-out (Theorem C.1) for $\lambda$ (ridge parameter) and $\sigma_y$ (parameters of $Y$ kernel):

$\lambda,\ \sigma_y = \arg\min \sum_{i=1}^M \dfrac{\left\| \psi(z_i) - K_{y_i Y}(K_{YY} + \lambda I)^{-1} K_{Z\cdot} \right\|_{\mathcal{H}_z}^2}{\left(1 - \left(K_{YY}(K_{YY} + \lambda I)^{-1}\right)_{ii}\right)^2}$

$W_1 = (K_{YY} + \lambda I)^{-1}$, $W_2 = W_1 K_{ZZ} W_1$
**Every $L$ mini-batches**
Sample $D_0$ RFF $R(\cdot)$
$W_1^r = R(Y)^\top W_1 R(Z)$, $W_2^r = R(Z)^\top W_2 R(Z)$
**Mini-batch**
Use $D$ random RFF out of $D_0$
Compute $R(y), R(z)$ (mini-batch)
$\hat{K}^c = K_{yy} \odot \left( K_{zz} - R(y)W_1^r R(z)^\top - \left(R(y)W_1^r R(z)^\top\right)^\top + R(y)W_2^r R(y)^\top \right)$
$\text{CIRCE} = \frac{1}{B(B-1)} \text{Tr}\left( H K_{xx} H \hat{K}^c \right),\ H = I - \frac{1}{B} 1_B 1_B^\top$

---

| Parameter | Values | |
|---|---|---|
| | CIRCE and HSCIC | GCM |
| conditional independence $\gamma$ | log space between $[1, 10^4]$; | log space between $[10^{-2}, 10^{-0.5}]$ |
| ridge regression $\lambda$ | { 0.001, 0.01, 0.1, 1 } | |
| RBF scale | { 0.001, 0.01, 0.1, 1 } | |

Table 2: Hyperparameters for CIRCE, HSCIC and GCM on synthetic datasets.

### E.1 UNIVARIATE CASES

Structural causal model for univariate case 1:

$$
\begin{aligned}
Y, \epsilon_Z &\sim \mathcal{N}(0, 1) \\
\epsilon_A, \epsilon_B &\sim \mathcal{N}(0, 0.1) \\
Z &= Y^2 + \epsilon_Z \\
A &= 0.5 Z \epsilon_A + 2Y \\
B &= 0.5 \exp\left(-AY\right) \sin(2AY) + 5Z + 0.2\epsilon_B
\end{aligned}
$$

Structural causal model for univariate case 2:

$$
\begin{aligned}
Y, \epsilon_Z &\sim \mathcal{N}(0, 1) \\
\epsilon_A, \epsilon_B &\sim \mathcal{N}(0, 0.1) \\
Z &= Y^2 + \epsilon_Z \\
A &= \exp(-0.5 Z^2) \sin 2Z + 2Y + 0.2\epsilon_A \\
B &= \sin(2AY) \exp(-0.5AY) + 5Z + 0.2\epsilon_B
\end{aligned}
$$

### E.2 MULTIVARIATE CASES

Structural causal model for multivariate case 1:

$$Y, \epsilon_{Z_i} \sim \mathcal{N}(0, 1)$$
$$\epsilon_A, \epsilon_B \sim \mathcal{N}(0, 0.1)$$
$$Z_i = Y^2 + \epsilon_{Z_i}$$
$$A = \exp(-0.5Z_1) + \sum_i Z_i \sin(Y) + 0.1\epsilon_A$$
$$B = \exp(-0.5Z_2)(\sum_i Z_i) + AY + 0.1\epsilon_B$$

Structural causal model for multivariate case 2:

$$Y_i, \epsilon_Z \sim \mathcal{N}(0, 1)$$
$$\epsilon_A, \epsilon_B \sim \mathcal{N}(0, 0.1)$$
$$Z = Y^T Y + \epsilon_Z$$
$$A = \exp(-0.5Z) + \sin \sum_i Y_i Z + 0.1\epsilon_A$$
$$B = \exp(-0.5Z)Z + \sum_i Y_i + Z + AY_1 + 0.1\epsilon_B$$

## F IMAGE DATA DETAILS

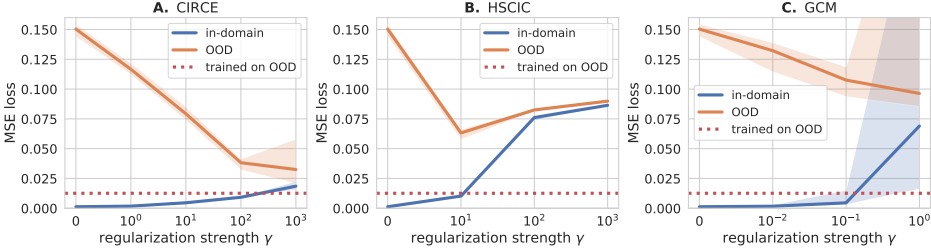

Figure 7: dSprites with nonlinear dependence. CIRCE used holdout data in training. Blue: in-domain test loss; orange: out-of-domain loss (OOD); red: loss for OOD-trained encoder. Solid lines: median over 10 seeds; shaded areas: min/max values.

For both dSpritres and Yale-B, we choose the following training hyperparameters over the validation set and *without* regularization: weight decay (1e-4, 1e-2), learning rate (1e-4, 1e-3, 1e-2) and length of training (200 or 500 epochs). These parameters are used for all runs (including the regularized ones). For dSprites the batch size was 1024. For Yale-B the batch size was 256. The results for both standard (Corollary C.5) and centered (Corollary C.6) CIRCE estimators were similar for dSprites (the reported one is standard), but the centered version was more stable for Yale-B (the reported one is centered). This is likely due to the bias arising from conditional expectation estimation. For dSprites, the training set contained 589824 points, and the holdout set size was 5898 points. For Yale-B, the training set contained 11405 points, and the holdout set size was 1267 points.

All kernels were Gaussian: $k(x, x') = \exp(-\|x - x'\|^2/(2\sigma^2))$. For $Y$, $\sigma^2$ from $[1.0, 0.1, 0.01, 0.001]$ and ridge regression parameter $\lambda$ from $[0.01, 0.1, 1.0, 10.0, 100.0]$. The other two kernels had $\sigma^2 = 0.01$ for linear and y-cone dependencies; for the nonlinear case, the kernel over $Z$ had $\sigma^2 = 1$ due to a different scaling of the distractor in that case.

We additionally tested a setting in which the $M$ holdout points used for conditional expectation estimation are not removed from the training data for CIRCE. As shown in Figure 7 for dSprites with non-linear dependence, this has little effect on the performance.

