# OpenReview forum: "Efficient Conditionally Invariant Representation Learning"
_ICLR.cc/2023/Conference — ICLR 2023 notable top 5%_

### Official Review · Reviewer_FXBx · 2022-10-22

**Confidence:** 3
**Correctness:** 4
**Technical Novelty And Significance:** 4
**Empirical Novelty And Significance:** 3
**Recommendation:** 8

**Clarity, Quality, Novelty And Reproducibility:**

The paper is overall well-written. There are adequate citations, and the originality of the work is clear. I think the results are not difficult to be reproduced. The authors even provide code.
There are a few unclear details though:
- The correspondence between the variables $X$, $Y$, $Z$, etc. and the actual attributes of the data in experiments except for the Extended Yale-B.
- It would be nice to emphasize that Eqs. (14-15) are not a structural equation model or anything that indicates causal directions.
- Clear definitions of some math symbols such as $\mu_{ZY \mid Y}$ and $K_XX$ are missing in the main part of the paper. (They might be in the appendices, but I did not check that carefully.)
- In p.5, Remark says the rate is minimax optimal for the conditional expectation but not for the entire CIRCE estimation, and there might be another approach that can improve the rate, am I right?

**Strength And Weaknesses:**

# Strengths
- The authors point out that Eq. (4) has a nice property that is useful for mini-batch training.
- They provide a clever way of estimating it using RHKSes.
- They prove the correctness of the proposed method.
- The experiments are interesting and demonstrating the effectiveness of the proposed method.

# Weakness
- I suppose that the paper claims the proposed method is computationally efficient, but I could not find any analysis or empirical results supporting it. In fact, Theorem 2.7 suggests that we want M to be large enough to compensate the slow rate, but calculating the kernel matrices may not be scalable in $M$.

**Summary Of The Paper:**

This paper proposes a measure of conditional independence called CIRCE that can be used as a regularizer in a large-scale mini-batch training. The key idea of CIRCE is that the conditional independence holds if and only if Eq. (4) holds, so we can measure the conditional dependence by how much the equation is violated. However, the equation must be checked for all functions g and h of the $L^2$ spaces, which is not directly tractable. The authors rewrite this condition by replacing the $L^2$ spaces with RKHSes, which reduces the equation to the norm of one single operator being zero (Eq. (10)). The estimation of the operator can be decomposed into the one related to the encoder and the other that does not depend on the encoder but requires a conditional expectation estimation. Conveniently, the regression part can be done beforehand and does not have to be calculated for each mini-batch. The authors prove the consistency with a rate of convergence of the proposed estimator and provide experiments on learning causal relationships using synthetic data and real image data.

**Summary Of The Review:**

The authors made great contributions to incorporating a conditional independence regularization in mini-batch training with a novel and interesting approach. They prove correctness in theory and confirm the effectiveness by experiments and provides many insights. I suggest accepting this paper.

---

> ### Author Response · Authors · 2022-11-14
> **Authors' response**
>
> Thank you for the review!
>
> > In p.5, Remark says the rate is minimax optimal for the conditional expectation but not for the entire CIRCE estimation, and there might be another approach that can improve the rate, am I right?
>
> This is an interesting topic, the minimax rate for CIRCE is indeed unknown. However we can set the holdout set size $M=B^2$, getting the standard $1/\sqrt{B}$ rate for a covariance operator estimate.
>
> > Theorem 2.7 suggests that we want M to be large enough to compensate the slow rate, but calculating the kernel matrices may not be scalable in M.
>
> Classical  methods for computing  kernel ridge regression indeed scale poorly (cubic in sample size), however this can easily be addressed. In appendix D, we discussed random Fourier features as one way to improve scaling with holdout sample size. More advanced scaling approaches for kernel methods such as FALKON [Rudi et al., 2017]  can scale kernel ridge regression up to millions of points. We will add a discussion on the scalability of our method.
>
>
>
> **General update**: we have modified the paper according to reviewer suggestions, highlighting the major changes in red. We will further work on clarifying the notation. We have also fixed an implementation bug for CIRCE that affected Figs. 6-7. Notably, our fixed code **improved our performance** for nonlinear dSprites.
> To clarify our error, we had accidentally swapped $K_{YY}$ in Eq. 12 with $K_{ZZ}$. The code error only applied in certain nonlinear experiments, but not the linear ones (e.g. linear dSprites). We re-ran all relevant experiments, and figures in the current paper reflect the performance of the corrected code.
>
> [Rudi et al., 2017] Rudi, A., Carratino, L., and Rosasco, L. (2017). Falkon: An optimal large scale kernel method.

---

### Official Review · Reviewer_ZZGR · 2022-10-23

**Confidence:** 3
**Correctness:** 3
**Technical Novelty And Significance:** 2
**Empirical Novelty And Significance:** 3
**Recommendation:** 6

**Clarity, Quality, Novelty And Reproducibility:**

The paper falls into a typical kernel based measure paper: propose a new operator, show that it could enjoy certain statistical property with properly chosen kernels, and finally provide a sample estimate with theoretical guarantee. I have a few questions:

- This paper uses the Eq. (3) and (4) (also a result from Daudin 1980) to derive a new CI measure and statistic. The benefit is that it could avoid the minibatch computation involving  $Z$ and $Y$.  It seems that the data are typically huge, so such computation may still need to be done in a mini-batch way?
- why *“the minibatch data on every minibatch in gradient descent requires impractically many samples* and can you show it in more details？ Note that the proposed method also invlolves minibatch estimate.
- Suggest to give a more detailed description (like math formulas) on the difference between the proposed method and previous kernel based statistic based on Eq. (2) like Zhang et al., 2011. This is important for readers to get the novel part.

- this method requires some hold-out data, but these data, removed from training data, would affect the training performance. Please some analysis on this point, or some quantitive results to show its effect on the final performance.
- In many cases like invariant learning, $Z$  is low dimensional, so perhaps avoiding the minibatcch computation of $Z$ and $Y$ does not matter much. From the experiment in Figure 1, it seems that when $d$ increases, the performance of CIRCE is outperformed by HSCISC, which seem to validate my conjecture? (but from Figure 2, the observation is opposite. Can you also explain on this?)

- Before the experiment part, I did not see any intuition that the method could perform better than previous methods. Can you provide any intuition or rough idea on this point in the main text?
- Writing: several notations are not defined in the main text. In section 2.3, $K_{xx}$, $\sigma_y$, $F_s$ , etc. in algorithm 1 have not been defined.



**Strength And Weaknesses:**

Pros:

- a new CI meansure with finite sample estimate with convergence guarantee
- the method is shown useful in some experiments
- writing is good (expect for several places where notations are not defined in the main text)

Cons:

- except for avoiding minibatch computing of $Z$ and $Y$, it is not clear what other benefits the proposed statistic could provide.
- the experiments are somewhat toy and the results are mixed

**Summary Of The Paper:**

This paper introduces a new measure of conditional independence for multivariate continuous variables, based on the equations of  Daudin, 1980. Although previous tests have also used other equivalent forms of equations from (Daudin, 1980), the proposed  CI measure and statistic seem to enjoy the advantage of avoiding minibatch computation involving $Z$ and $Y$. Finite sample estimate with convergence guarantees and strategies for efficient estimation from data are also provided.

**Summary Of The Review:**

I'm kind of between weak rej and wek acc. The method shows some usefulness in experiments and has good theoretic characterizations. On the other hand, it does not provide an intuitive sense of why it could be better. I wait for authors response before recommending an acceptance.

---

> ### Author Response · Authors · 2022-11-14
> **Authors' response (part 1/2)**
>
> Thank you for the review! Below we address the questions and concerns raised.
>
> > Except for avoiding minibatch computing of Z and Y, it is not clear what other benefits the proposed statistic could provide.
>
> The main benefit of our method is that we avoid additional regressions from $Y$ to $\varphi(x)$, where the (deep NN) features $\varphi(x)$ are updated and adapted using stochastic gradient descent.
> By contrast with CIRCE, earlier measures of conditional dependence require
> regressing on each minibatch from $Y$ towards  features $\varphi(x)$ that change at each SGD step. This is expensive computationally, and has high variance for common minibatch sizes (which leads to inaccurate enforcement of the conditional dependence restriction). Circe does still require regressing from $Y$ to features  $\psi(Z)$, however the features  $\psi(Z)$ of $Z$ are static, and this regression can be accomplished offline, with low variance,  using a large number of datapoints.
>
> Offline computation of the regression from $Y$ to features  $\psi(Z)$ has several advantages:
> it can be done on a CPU without VRAM limitations of a GPU; and it can be scaled up with either random Fourier features (discussed in appendix D) or more advanced scaling approches for kernel methods such as FALKON [Rudi et al., 2017]  (which scales up to millions of points).
>
> > The experiments are somewhat toy and the results are mixed
>
> We emphasize that the experiments are chosen to demonstrate that CIRCE performes in accordance with claims made, notably when $X$ is high dimensional and requires training sophisticated NN features. We also underline that the non-linear dSprites result suffered from a bug in our initial code (described at the end of this response); the correct implementation improved the performance (see updated submission), showing that CIRCE significantly outperforms HSCIC and GCM in this setting, thus validating our claims regarding the benefits of CIRCE.
>
> > The benefit is that it could avoid the minibatch computation involving Z and Y. It seems that the data are typically huge, so such computation may still need to be done in a mini-batch way?
>
> Please see our reply above. First, the benefit is not just avoiding the regression of $\psi(Z)$ and Y in minibatch, but also removing the $Y$ to $\varphi(X)$ regression altogether. Second, scalable regression algorithm exists for scaling up to millions of points in the batch setting when regressing $Y\rightarrow \psi(Z)$.
>
> > Suggest to give a more detailed description (like math formulas) on the difference between the proposed method and previous kernel based statistic based on Eq. (2) like Zhang et al., 2011. This is important for readers to get the novel part.
>
> We have now added these expressions and an explanation of the difference in the revised document.
>
> > Effects of held-out data on training performance.
>
> We appreciate this suggestion, and have run additional experiments on dSprites without removing the holdout data from the training set, showed in Fig. 8 (page 20) of the updated paper. Interestingly, we did not see a significant change in performance for CIRCE. Thus, while it's (statistically speaking) necessary to split the data, a pragmatic approach for practitioners, in a limited data scenario, would be not to use a split, since performance is almost identical.
>
> We conjecture that this is because the holdout data constitutes a small part of the total dataset. Thus, in each minibatch, only a few points come from the holdout dataset, resulting in a small bias in the estimator.
>
> > From the experiment in Figure 1, it seems that when d increases, the performance of CIRCE is outperformed by HSCISC, which seem to validate my conjecture? (but from Figure 2, the observation is opposite. Can you also explain on this?)
>
> Multivariate case 1 has high dimensional $Z$, while case 2 has high dimensional $Y$. The latter case presents the greater challenge when regressing $Y \to \psi(Z)$ (especially on minibatches), as is confirmed experimentally. Figure 2 shows that HSCIC and CIRCE have very similar behaviour as output dimension changes.
> By contrast, in Figure 3, we expect HSCIC to do worse for increasing input dimension $d$. We have confirmed this by adding additional experiments to  Figure 3, where we increase input dimension $d$ up to 20. HSCIC's performance degrades with increasing dimensionality until $d=20$ where it completely fails. We updated this section in the paper.
>
> [Rudi et al., 2017] Rudi, A., Carratino, L., and Rosasco, L. (2017). Falkon: An optimal large scale kernel method.

---

> ### Author Response · Authors · 2022-11-14
> **Authors' response (part 2/2)**
>
> > In many cases like invariant learning, $Z$ is low dimensional, so perhaps avoiding the minibatch computation of $Z$ and $Y$ does not matter much
>
> It's correct (as we've seen in experiments) that the advantage of CIRCE is greater when the dimension of $Y$ (the input to the regression) increases. However, we also anticipate a benefit when the relation between $Y$ and $\psi(Z)$ is very non-smooth, hence poorly learned on a minibatch.  This is in accordance with the theory on consistency for the conditional mean embedding (CME). Note also that the CME converges more slowly ($M^{-1/4}$) than the feature covariance ($B^{-1/2}$), meaning that we will benefit correspondingly more by regression from $Y$ to $\psi(Z)$ on a larger batch.
>
>
> > Intuition as to why it could perform better.
>
> Please see earlier comments.
> Note (in addition to these earlier remarks) that unlike GCM, we consider a rich set of features for the distractor $Z$, and thus we are able to capture high order correlations (Explained in Sec. 4.2.1, non-linear example).
>
> > Missing notations.
> We will add the missing notation in our final version.
>
>
> **General update**: we have modified the paper according to reviewer suggestions, highlighting the major changes in red. We will further work on clarifying the notation. We have also fixed an implementation bug for CIRCE that affected Figs. 6-7. Notably, our fixed code **improved our performance** for nonlinear dSprites.
> To clarify our error, we had accidentally swapped $K_{YY}$ in Eq. 12 with $K_{ZZ}$. The code error only applied in certain nonlinear experiments, but not the linear ones (e.g. linear dSprites). We re-ran all relevant experiments, and figures in the current paper reflect the performance of the corrected code.
>
> We hope we have addressed your concerns in the above -- if so, we would be grateful if you could consider raising the review score. If any questions or concerns remain, then please let us know, and we will do our best to answer them.

---

> > ### Comment · Reviewer_ZZGR · 2022-11-17
> > **Thanks for response and clarification**
> >
> > Thanks very much for the response and additional experiments to validate the proposed method. I have increased my score accordingly.

---

### Official Review · Reviewer_QXAF · 2022-10-24

**Confidence:** 3
**Correctness:** 4
**Technical Novelty And Significance:** 3
**Empirical Novelty And Significance:** 4
**Recommendation:** 8

**Clarity, Quality, Novelty And Reproducibility:**

I find this paper very clearly written. As far as I know, the proposed method is novel and is good for solving many realistic problems where  $X$ and $(Y,Z)$ need to be treated different in computation. I didn't check the details of the proofs and experiments, but I tend to believe they are correct and reproducible.

**Strength And Weaknesses:**

Learning conditionally invariant representation is a very important problem that has the potential to solve a wide of issues in machine learning, such as distribution shift, spurious correlation, racial bias from datasets, etc. The authors consider a quite general setup where one seeks to convert the hard-to-measure conditional independence into a manageable form.

The major strength of this paper is a new measure that can be efficiently computed from data. This measure involves marginal independence between $X$ and $(Y,Z)$ under square-integrable functions, thus cleverly separating the computation of (potentially high-dimensional) inputs $X$ from $(Y,Z)$.

Also, the empirical evaluation seems to favor this CIRCE method when compared with existing methods. This suggests that the proposed approach has good potential impact.

A weakness of this paper, partly due to the limit of the paper length, is a lack of analysis of the statistical efficiency (or power in statistical tests).

**Summary Of The Paper:**

In this paper, the authors study learning conditionally invariant representation of inputs $X$ such that conditional on label $Y$, the representation $\varphi(X)$ and the distractor $Z$ are independent. This problem is motivated by broad applications in fairness, domain invariant learning, causal representation learning, etc.

A challenge in enforcing conditional independence is that it is hard to measure conditional independence, effectively reducing the sample size if we naively split the data according to values of $Y$. A major contribution of this paper is reducing conditionally independent representations to marginally independent representations, based on a result of [1]. Then the authors proceed, by invoking reproducing kernel Hilbert space, to transform the problem into an equivalent statement (Thm 2.5) in which we look for a CIRCE operator such that its Hilbert-Schmidt norm is zero. On finite-sample data, an estimator of the CIRCE operator is given, so this effectively becomes a regularizer.

The authors then present several numerical simulations and data analysis, demonstrating the advantages of the proposed CIRCE method with existing approaches in the literature.




[1] JJ Daudin. Partial association measures and an application to qualitative regression.

**Summary Of The Review:**

The authors proposed a new measure called CIRCE for enforcing conditional independence, and this measure can be efficiently computed from data. Overall I believe this is a good paper, and thus I recommend acceptance of this paper.

---

> ### Author Response · Authors · 2022-11-14
> **Authors' response**
>
> Thank you for the review!
>
> > Lack of analysis of the statistical efficiency (or power in statistical tests)
>
> Regarding the use of CIRCE in a statistical test: this is definitely a useful direction to pursue for future work. We believe we could use similar techniques as in [Zhang et al., 2011] to obtain a test threshold. In the event that adaptive neural net features are used on $X$, it might be helpful to optimize these to maximize (a proxy for) test power on held-out data, as in [Liu et al., 2020].
>
>
> **General update**: we have modified the paper according to reviewer suggestions, highlighting the major changes in red. We will further work on clarifying the notation. We have also fixed an implementation bug for CIRCE that affected Figs. 6-7. Notably, our fixed code **improved our performance** for nonlinear dSprites.
> To clarify our error, we had accidentally swapped $K_{YY}$ in Eq. 12 with $K_{ZZ}$. The code error only applied in certain nonlinear experiments, but not the linear ones (e.g. linear dsprites). We re-ran all relevant experiments, and figures in the current paper reflect the performance of the corrected code.
>
> [Zhang et al., 2011] Zhang, K., Peters, J., Janzing, D., and Scholkopf, B. (2011). Kernel-based conditional independence test
> and application in causal discovery
>
> [Liu et al., 2020] Liu, F., Xu, W., Lu, J., Zhang, G., Gretton, A., and Sutherland, D. J. (2020). Learning deep kernels for non-
> parametric two-sample tests

---

### Decision · Program_Chairs · 2023-01-20

**Decision:**

Accept: notable-top-5%

**Justification For Why Not Higher Score:**

N/A

**Justification For Why Not Lower Score:**

The method is novel, theoretically justified and for a problem of broad interest in many domains.

**Metareview: Summary, Strengths And Weaknesses:**

The paper introduces a novel method for representation learning that is independent of an observed environment/distractor variable Z, conditional on the label Y. Such invariant representation learning finds applications in fairness, out-of-distribution generalization and causal representation learning. The paper proposes a mathematically rigorous kernel based method that is computationally tractable and can handle arbitrary non-linear relationships and leads to good empirical performance.

**Note From Pc:**

if the above contains the word "oral" or "spotlight" please see: "oral" presentation means -> notable-top-5% and "spotlight" means -> notable-top-25%. As stated in our emails, we are disassociating presentation type from AC recommendations